# BYZANTINE ROBUST COOPERATIVE MULTI-AGENT REINFORCEMENT LEARNING AS A BAYESIAN GAME

**Simin Li[1], Jun Guo[1], Jingqiao Xiu[1], Ruixiao Xu[1], Xin Yu[1], Jiakai Wang[2],**
**Aishan Liu[1,4], Yaodong Yang[3] \*, Xianglong Liu[1,2,4] \***
[1]SKLSDE Lab, Beihang University, China    [2]Zhongguancun Laboratory, China
[3]Institute of Artificial Intelligence, Peking University & BigAI, China
[4]Institute of data space, Hefei Comprehensive National Science Center, China

## ABSTRACT

In this study, we explore the robustness of cooperative multi-agent reinforcement learning (c-MARL) against Byzantine failures, where any agent can enact arbitrary, worst-case actions due to malfunction or adversarial attack. To address the uncertainty that any agent can be adversarial, we propose a Bayesian Adversarial Robust Dec-POMDP (BARDec-POMDP) framework, which views Byzantine adversaries as nature-dictated types, represented by a separate transition. This allows agents to learn policies grounded on their posterior beliefs about the type of other agents, fostering collaboration with identified allies and minimizing vulnerability to adversarial manipulation. We define the optimal solution to the BARDec-POMDP as an *ex interim* robust Markov perfect Bayesian equilibrium, which we proof to exist and the corresponding policy weakly dominates previous approaches as time goes to infinity. To realize this equilibrium, we put forward a two-timescale actor-critic algorithm with almost sure convergence under specific conditions. Experiments on matrix game, Level-based Foraging and StarCraft II indicate that, our method successfully acquires intricate micromanagement skills and adaptively aligns with allies under worst-case perturbations, showing resilience against non-oblivious adversaries, random allies, observation-based attacks, and transfer-based attacks.

## 1 INTRODUCTION

Cooperative multi-agent reinforcement learning (c-MARL) (Rashid et al., 2018; Yu et al., 2021; Kuba et al., 2021) has shown remarkable efficacy in managing groups of agents with aligned interests in complex tasks (Vinyals et al., 2019; Berner et al., 2019). Nevertheless, real-world applications often deviates from the presumption of full cooperation. In robot swarm control (Hüttenrauch et al., 2019), individual robots may act unpredictably due to hardware or software malfunctions, or even display worst-case adversarial actions if compromised by a *non-oblivious* adversary (Gleave et al., 2019; Lin et al., 2020; Dinh et al., 2023; Liu et al., 2019; 2020a;b; 2023; Wang et al., 2021). Such *uncertainty of allies* undermine the cooperative premise of c-MARL, rendering the learned policy non-robust.

In single-agent reinforcement learning (RL), robustness under uncertainty is addressed through a maximin optimization between an uncertainty set and a robust agent within the framework of robust Markov Decision Processes (MDPs) (Nilim & El Ghaoui, 2005; Iyengar, 2005; Wiesemann et al., 2013; Pinto et al., 2017; Tessler et al., 2019; Zhang et al., 2020a). However, ensuring robustness in c-MARL when dealing with uncertain allies presents a greater challenge. This is largely due to the potential for *Byzantine failure* (Yin et al., 2018; Xue et al., 2021), situations where defenders are left in the dark regarding which ally may be compromised and what their resulting actions might be.

To address Byzantine failures, we employ a Bayesian game approach, which treats Byzantine adversaries as *types* assigned by nature, with each agent operating unaware of others' type. We formalize robust c-MARL as a Bayesian Adversarial Robust Dec-POMDP (BARDec-POMDP), where existing robust MARL researches (Li et al., 2019; Sun et al., 2022; Phan et al., 2020; 2021) can be reinterpreted as pursuing an *ex ante* equilibrium (Shoham & Leyton-Brown, 2008), viewing

---

*Corresponding Authors. E-mails: yaodong.yang@pku.edu.cn, xlliu@buaa.edu.cn.

all other agents as potential adversaries. However, these methods might not yield optimal outcomes as they can mask the trade-offs between the equilibria that cooperative and robustness-focused agents aim for. Moreover, this approach can result in overly conservative strategies (Li et al., 2019; Sun et al., 2022), given the low likelihood of adversaries taking control of all agents.

Instead, we seek an *ex interim* mixed-strategy robust Markov perfect Bayesian equilibrium, which weakly dominates the policy of *ex ante* equilibrium in previous robust MARL studies as time goes to infinity. Agents in our *ex interim* equilibrium makes decisions based on its inferred posterior belief over other agents, enhancing cooperation with allies and defense against adversaries concurrently. To realize this equilibrium, we derive a robust Harsanyi-Bellman equation for value function update and introduce a two-timescale actor-critic algorithm, with almost sure convergence under certain assumptions. Experiments in matrix game, Level-based Foraging and StarCraft II shows our defense exhibits intricate micromanagement skills and adaptively aligns with allies under worst-case perturbations. Consequently, our defense outperforms existing baselines under non-oblivious adversaries, random allies, observation-based attacks and transfer-based attacks by large margins.

**Contribution.** Our contributions are two-fold: first, we theoretically formulate Byzantine adversaries in c-MARL as a BARDec-POMDP, and concurrently pursues robustness and cooperation by targeting an *ex interim* equilibrium. Secondly, to achieve this equilibrium, we devise an actor-critic algorithm that ensures almost sure convergence under certain conditions. Empirically, our method exhibits greater resilience against a broad spectrum of adversaries on three c-MARL environments.

**Related Work.** Our research belongs to the field of robust RL, theoretically framed as robust MDPs (Nilim & El Ghaoui, 2005; Iyengar, 2005; Tamar et al., 2013; Wiesemann et al., 2013). This framework trains a defender to counteract a worst-case adversary amid uncertainty, which can stem from environment transitions (Pinto et al., 2017; Mankowitz et al., 2019), actions (Tessler et al., 2019), states (Zhang et al., 2020a; 2021) and rewards (Wang et al., 2020). In robust MARL, action uncertainty has been a central focus. M3DDPG (Li et al., 2019) enhances robustness in MARL through agents taking jointly worst-case actions under a small perturbation budget. Evaluation was done via one agent consistently introducing worst-case perturbations. This is later known as *adversarial policy* (Gleave et al., 2019) or *non-oblivious adversary* (Dinh et al., 2023), a practical and detrimental form of attack. Follow-up works either enhanced M3DDPG (Sun et al., 2022) or defended against uncertain adversaries by presupposing each agent as potentially adversarial (Nisioti et al., 2021; Phan et al., 2020; 2021), which our BARDec-POMDP formulation interprets as seeking a conservative *ex ante* equilibrium. Another approach by Kalogiannis et al. (2022) studies a special case that the adversary is known. Besides action perturbation, studies have also explored robust MARL under uncertainties in reward (Zhang et al., 2020c), environmental dynamics (Zhao et al., 2020), and observations (Han et al., 2022; He et al., 2023; Zhou & Liu, 2023).

Bayesian games and their MARL applications represents another relevant field. With roots in Harsanyi's pioneering work (Harsanyi, 1967), Bayesian games have been used to analyze games with incomplete information by transforming them into complete information games featuring chance moves made by nature. Within MARL, Bayesian games have been utilized to coordinate varying agent types, a concept known as ad hoc coordination (Albrecht & Ramamoorthy, 2015; Albrecht et al., 2016; Stone et al., 2010; Barrett et al., 2017). This problem was theoretically framed as a stochastic Bayesian game and solved using the Harsanyi-Bellman ad hoc coordination algorithm (Albrecht & Ramamoorthy, 2015). Subsequent research has concentrated on agents with varying types (Ravula, 2019), open ad hoc teamwork (Rahman et al., 2022), and human coordination (Tylkin et al., 2021; Strouse et al., 2021). Our work differs from these works by assuming a worst-case, non-oblivious adversary with conflicting goals, whereas in ad hoc coordination, agents have common goals and non-conflicting secondary objectives (Grosz & Kraus, 1999; Mirsky et al., 2022).

## 2 PROBLEM FORMULATION

### 2.1 COOPERATIVE MARL AND ITS FORMULATION

The problem of c-MARL can be formulated as a Decentralized Partially Observable Markov Decision Process (Dec-POMDP) (Oliehoek & Amato, 2016), defined as a tuple:

$$\mathcal{G} := \langle \mathcal{N}, \mathcal{S}, \mathcal{O}, O, \mathcal{A}, \mathcal{P}, R, \gamma \rangle,$$

where $\mathcal{N} = \{1, ..., N\}$ is the set of $N$ agents, $\mathcal{S}$ is the global state space, $\mathcal{O} = \times_{i \in \mathcal{N}} \mathcal{O}^i$ is the observation space, with $O$ the observation emission function. $\mathcal{A} = \times_{i \in \mathcal{N}} \mathcal{A}^i$ is the joint action space, $\mathcal{P} : \mathcal{S} \times \mathcal{A} \to \Delta(\mathcal{S})$ is the state transition probability, mapping from current state and joint actions to a probability distribution over the state space. $R : \mathcal{S} \times \mathcal{A} \to \mathbb{R}$ is the shared reward function for cooperative agents and $\gamma \in [0, 1)$ is the discount factor.

At time $t$ and global state $s_t \in \mathcal{S}$, each agent $i$ adds current observation $o_t^i$ to its history and gets $H_t^i = [o_0^i, a_0^i, ...o_t^i]$. Then, each agent $i$ selects its action $a_t^i \in \mathcal{A}^i$ using its policy $\pi^i(\cdot|H_t^i)$, which maps current history to its action space. The global state then transitions to $s_{t+1}$ according to transition probability $\mathcal{P}(s_{t+1}|s_t, \mathbf{a}_t)$, with $\mathbf{a}_t = \{a_t^1, ..., a_t^N\}$ the joint actions. Each agent receives a *shared* reward $r_t = R(s_t, \mathbf{a}_t)$. The goal of all agents is to learn a joint policy $\pi = \prod_{i \in \mathcal{N}} \pi^i$ that maximize the long-term return $J(\pi) = \mathbb{E}\left[\sum_{t=0}^{\infty} \gamma^t r_t | s_0, \mathbf{a}_t \sim \pi(\cdot|H_t)\right]$.

## 2.2 BAYESIAN ADVERSARIAL ROBUST DEC-POMDP

In numerous real-world situations, some allies may experience Byzantine failure (Yin et al., 2018; Xue et al., 2021) and thus, not perform cooperative actions as expected. This includes random actions due to hardware/software error and adversarial actions if being controlled by an adversary, which violates the fully cooperative assumption in Dec-POMDP. We propose *Bayesian Adversarial Robust Dec-POMDP (BARDec-POMDP)* to cope with uncertainties in agent actions, defined as follows:

$$\hat{\mathcal{G}} := \langle \mathcal{N}, \mathcal{S}, \Theta, \mathcal{O}, O, \mathcal{A}, \mathcal{P}^\alpha, \mathcal{P}, R, \gamma \rangle,$$

where $\mathcal{N}$, $\mathcal{S}$, $\mathcal{O}$, $O$, $\mathcal{A}$, $\gamma$ represent the number of agents, global state space, observation space, observation emission function, joint action space and discount factor, following Dec-POMDP.

As depicted in Fig. 1, BARDec-POMDP views the Byzantine adversary as an uncertain transition characterized by type $\theta$ and adversarial policy $\hat{\pi}$. At the start of each episode, a *type* $\theta$ is selected from the type space $\Theta = \times_{i \in \mathcal{N}} \Theta^i$, with $\theta^i = \{0, 1\}$. $\theta^i = 0$ indicates the agent is cooperative and $\Theta^i = 1$ signifies adversaries. At time $t$, if agent $i$ is assigned $\theta^i = 1$, the action $a_t^i$ taken by cooperative agent $i$ with policy $\pi^i(\cdot|H_t^i)$ is replaced by action $\hat{a}_t^i$ sampled from an adversary with policy $\hat{\pi}^i(\cdot|H_t^i, \theta)$. The attack process is characterized by action perturbation

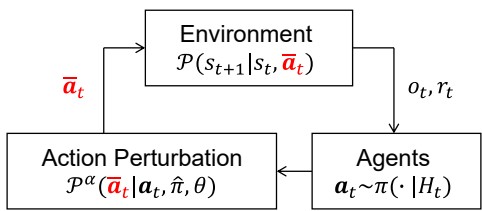

Figure 1: Framework of c-MARL with Byzantine adversaries. The action taken by agents with $\theta^i = 1$ are replaced by the adversary policy $\hat{\pi}^i$.

probability $\mathcal{P}^\alpha(\overline{\mathbf{a}}_t|\mathbf{a}_t, \hat{\pi}, \theta) = \prod_{i \in \mathcal{N}} \hat{\pi}^i(\cdot|H_t^i, \theta) \cdot \theta^i + \delta(\overline{a}_t^i - a_t^i) \cdot (1 - \theta^i)$ that maps joint actions $\mathbf{a}_t$ to joint actions with perturbations $\overline{\mathbf{a}}_t$, where $\delta(\cdot)$ is the Dirac delta function. Note that the actions of cooperative agents and adversaries are taken simultaneously. Finally, the state transition probability $\mathcal{P}(s_{t+1}|s_t, \overline{\mathbf{a}}_t)$ takes the perturbed actions and output the state of next timestep. The shared reward $r_t = R(s_t, \overline{\mathbf{a}}_t)$ for (cooperative) agents is defined over perturbed actions. Given type $\theta$, the value function can be defined as $V_\theta(s) = \mathbb{E}\left[\sum_{t=0}^{\infty} \gamma^t r_t | s_0 = s, \mathbf{a}_t \sim \pi(\cdot|H_t), \hat{\mathbf{a}}_t \sim \hat{\pi}(\cdot|H_t, \theta)\right]$. We leave the goal of adversary and robust agents to Section. 2.3 and 2.4 below.

Our BARDec-POMDP formulation is flexible and draws close connection with current literature. Regarding type space, Dec-POMDP can be viewed as a BARDec-POMDP with $\Theta = \mathbf{0}_N$. Robust MARL approaches, such as M3DDPG (Li et al., 2019) and ROMAX (Sun et al., 2022) assumes agents are entirely adversarial, which refers to type space $\Theta = \mathbf{1}_N$. Subsequent robust MARL researchs (Phan et al., 2020; 2021; Nisioti et al., 2021), though not explicitly defining the type space, can be integrated in our BARDec-POMDP. Our formulation also draws inspiration from state-adversarial MDP (Zhang et al., 2020a) which considers adversary as a part of decision making process, and probabilistic action robust MDP (Tessler et al., 2019) by their formulation of action perturbation.

## 2.3 THREAT MODEL

The robustness towards action perturbations in both single and multi-agent RL has gained prominence since the pioneering works of (Tessler et al., 2019; Li et al., 2019). Action uncertainties, formulated as a type of adversarial attack known as *adversarial policy* (Gleave et al., 2019; Wu et al., 2021; Guo et al., 2021), or *non-oblivious adversary* (Dinh et al., 2023), represent a pragmatic and destructive

form of attack that is challenging to counter. In line with these works, we propose a practical threat model with certain assumptions on attackers and defenders.

**Assumption 2.1** (Attacker's capability and limitations). At the onset of an episode, the attacker can select $\theta$ and *arbitrarily* manipulate the actions of agents with type $\theta^i = 1$. Within an episode, the type cannot be altered and we assume there is only one attacker.

Our main focus in this paper is to model action uncertainties of unknown agents in c-MARL as *types* shaped by nature and to advance corresponding solution concept. In line with (Li et al., 2019), we assume one agent is vulnerable to action perturbations in each episode. In real-world, the type space can be more complicated, potentially involving adversaries controlling multiple agents (Nisioti et al., 2021), perturbing actions intermittently (Lin et al., 2017), or featuring a non-binary type space (Xie et al., 2022). These variations can be viewed as straightforward extensions of our work.

**Proposition 2.1** (Existence of worst-case adversary). For any robust c-MARL with fixed agent policy, a worst-case (*i.e.*, most harmful) adversary exists.

*Proof sketch.* Since defender policies are fixed, they can be considered part of the environment transitions for attackers. Thus, the attackers solve an RL problem. See full proof in Appendix. A.1.

**Assumption 2.2** (Defender's capability and limitations). The defender can use all available information during training stage, including global state and information of other agents. However, during testing, the defender relies solely on partial observations and is agnostic of the type of other agents. The defender's policy is fixed during an attack and must resist the worst-case adversary $\hat{\pi}_*$.

## 2.4 SOLUTION CONCEPT

In this section, we first introduce the non-optimal solution concept seek by existing robust c-MARL methods, then pose our optimal solution concept for BARDec-POMDP. Specifically, existing robust c-MARL methods blindly maximize reward without considering the type of others. This is akin to an *ex ante* equilibrium in Bayesian game (Shoham & Leyton-Brown, 2008), where agents make decisions based on the *prior* belief about the types of other agents.

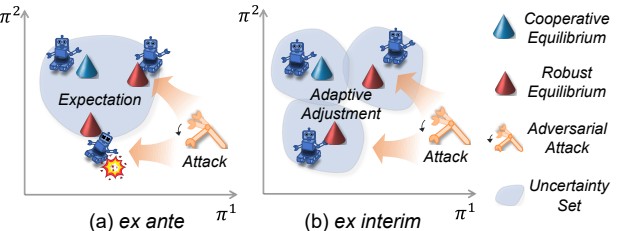

Figure 2: *ex ante* RMPBE obscures differences between each type by taking expectation, while our *ex interim* RMPBE adapts to current type.

**Definition 2.1** (*ex ante* robustness). A joint cooperative policy $\pi_*^{EA} = (\pi_*^{EA,i})_{i \in \mathcal{N}}$ and adversarial policy $\hat{\pi}_*^{EA} = (\hat{\pi}_*^{EA,i})_{i \in \mathcal{N}}$ forms an *ex ante* robust Markov perfect Bayesian equilibrium (RMPBE), if for all $p(\theta), s \in \mathcal{S}, H \in (\mathcal{O} \times \mathcal{A})^*$,

$$(\pi_*^{EA}(\cdot|H), \hat{\pi}_*^{EA}(\cdot|H,\theta)) \in \arg\max_{\pi(\cdot|H)} \mathbb{E}_{p(\theta)}\left[\min_{\hat{\pi}(\cdot|H,\theta)} V_\theta(s)\right], \tag{1}$$

with $V_\theta(s) = \sum_{\overline{\mathbf{a}} \in \mathcal{A}} \mathcal{P}^\alpha(\overline{\mathbf{a}}|\mathbf{a}, \hat{\pi}, \theta) \prod_{i \in \mathcal{N}} \pi^i(a^i|H^i)(R(s, \overline{\mathbf{a}}) + \gamma \sum_{s' \in \mathcal{S}} \mathcal{P}(s'|s, \overline{\mathbf{a}})V_\theta(s'))$.

By maximizing the expected value under prior $p(\theta)$ over types, as illustrated in Fig. 2, the policy might struggle to balance the different equilibrium corresponding to cooperation and robustness against different agents as (Byzantine) adversaries. In contrast, we propose a more refined *ex interim* robustness concept, such that under current history, each agent make optimal decisions to maximize their expected value from their posterior belief of current type, with following assumptions:

**Assumption 2.3.** Assume belief and policy are updated under following conditions: (1) *Consistency*. At each timestep $t$, each agent updates its belief $b_t^i = p(\theta|H_t^i)$ of the current type by Bayes' rule. (2) *Sequential rationality*. Each policy maximizes the expected value function under belief $b^i$.

Here we use $\pi^i(\cdot|H^i, b^i)$ to denote the explicit dependence on belief $b^i$. The two conditions are common in dynamic games with incomplete information (Shoham & Leyton-Brown, 2008).

**Definition 2.2** (*ex interim* robustness). Under Assumption 2.3 and let $b = (b^i)_{i \in \mathcal{N}}$, a joint cooperative policy $\pi_*^{EI} = (\pi_*^{EI,i})_{i \in \mathcal{N}}$ and adversarial policy $\hat{\pi}_*^{EI} = (\hat{\pi}_*^{EI,i})_{i \in \mathcal{N}}$ forms an *ex interim* robust Markov perfect Bayesian equilibrium, if $\forall s \in \mathcal{S}, H \in (\mathcal{O} \times \mathcal{A})^*$,

$$(\pi_*^{EI}(\cdot|H, b), \hat{\pi}_*^{EI}(\cdot|H, \theta)) \in \arg\max_{\pi(\cdot|H, b)} \mathbb{E}_{p(\theta|H)} \left[ \min_{\hat{\pi}(\cdot|H, \theta)} V_\theta(s)) \right]. \tag{2}$$

Note that both *ex ante* and *ex interim* RMPBE requires optimality of each individual agent $i$. To reduce notation complexity, we use joint policy and belief instead.

**Proposition 2.2** (Existence of RMPBE). Assume a BARDec-POMDP of finite agents, finite set of state, observation and action space, agents use stationary policies, the type space $\Theta$ is a compact set, then *ex ante* and *ex interim* mixed strategy robust Markov perfect Bayesian equilibrium exists.

*Proof sketch.* The proof is done by first showing the policy and its corresponding value function, with uncertainties of the current type and presence of the adversaries, satisfy the requirements of Kakutani's fixed point theorem. Next, by Kakutani's fixed point theorem, there always exists an optimal fixed point corresponding to a mixed strategy RMPBE. See full proof in Appendix. A.2.

Unlike c-MARL with optimal deterministic policies (Oliehoek et al., 2008), in robust c-MARL, a pure-strategy equilibrium is not guaranteed to exist. This is intuitive since zero-sum games do not always have a pure-strategy equilibrium. The finding suggests the optimal policies for robust c-MARL are stochastic. Next, we show the relation between *ex ante* and *ex interim* equilibrium.

**Proposition 2.3.** Under Assumption 2.3, given finite type space and the prior of each type is not zero, as $t \to \infty$, $\pi_*^{EI}(\cdot|H_t, b_t)$ weakly dominates $\pi_*^{EA}(\cdot|H_t)$ under the worst-case adversary.

*Proof sketch.* As $t \to \infty$, by the consistency of Bayes' rule, the belief converges to the true type. Thus, *ex interim* policies that maximize the value function under the true type are guaranteed to weakly dominate (*i.e.*, have value higher or equal to) *ex ante* policies. See full proof in Appendix A.3.

## 3 Algorithm

In this section, we explain how to find the optimal solution in Definition 2.2. We start by defining the robust Harsanyi-Bellman equation, an update rule of value function which converges to a fixed point. Then, we develop a two-timescale actor-critic algorithm that considers belief of others' type, which ensures almost sure convergence under assumptions in stochastic approximation theory.

### 3.1 Robust Harsanyi-Bellman Equation

We first define the Bellman-type update of value functions for our *ex interim* equilibrium. Considering the Q function before and after action perturbation, we can formulate the Q function via cumulative reward, with posterior belief $b^i = p(\theta|H^i)$ over type:

$$Q^i(s, \mathbf{a}, b^i) = \mathbb{E}_{p(\theta|H^i)} \left[ \mathbb{E} \left[ \sum_{t=0}^{\infty} \gamma^t r_t \Big| s_0 = s, \mathbf{a}_0 = \mathbf{a}, \mathbf{a}_t \sim \pi(\cdot|H_t, b_t), \hat{\mathbf{a}}_t \sim \hat{\pi}(\cdot|H_t, \theta) \right] \right], \tag{3}$$

$$Q^i(s, \overline{\mathbf{a}}, b^i) = \mathbb{E}_{p(\theta|H^i)} \left[ \mathbb{E} \left[ \sum_{t=0}^{\infty} \gamma^t r_t \Big| s_0 = s, \overline{\mathbf{a}}_0 = \overline{\mathbf{a}}, \mathbf{a}_t \sim \pi(\cdot|H_t, b_t), \hat{\mathbf{a}}_t \sim \hat{\pi}(\cdot|H_t, \theta) \right] \right], \tag{4}$$

The two Q functions are defined with different purpose. $Q^i(s, \mathbf{a}, b^i)$ is the *expected* Q function before action perturbation, suitable for decision making of defenders, such as fictitious self-play (Heinrich & Silver, 2016) and soft actor-critic (Haarnoja et al., 2018). In this way, the action perturbation can be viewed as part of the environment transition, resulting in $\overline{\mathcal{P}}(s'|s, \mathbf{a}, \hat{\pi}, \theta) = \mathcal{P}(s'|s, \overline{\mathbf{a}}) \cdot \mathcal{P}^\alpha(\overline{\mathbf{a}}|\mathbf{a}, \hat{\pi}, \theta)$.

On the other hand, $Q^i(s, \overline{\mathbf{a}}, b^i)$ is the Q function with actions *taken* by the adversary, suitable for policy gradients (Sutton & Barto, 2018) and decision making of the adversary. For $Q^i(s, \overline{\mathbf{a}}, H^i)$, the action perturbation is integrated into the policy of robust agents, resulting in a mixed policy $\overline{\pi}(\hat{\mathbf{a}}|H, b, \theta) = \mathcal{P}^\alpha(\overline{\mathbf{a}}|\mathbf{a}, \hat{\pi}, \theta) \cdot \pi(\mathbf{a}|H, b) = (1 - \theta) \cdot \pi(\mathbf{a}|H, b) + \theta \cdot \hat{\pi}(\hat{\mathbf{a}}|H, \theta)$. The relationship between the two Q functions is as follows:

$$Q^i(s, \mathbf{a}, b^i) = \sum_{\theta \in \Theta} p(\theta|H^i) \sum_{\hat{\mathbf{a}} \in \mathcal{A}} \mathcal{P}^\alpha(\overline{\mathbf{a}}_t|\mathbf{a}_t, \hat{\pi}, \theta) Q^i(s, \overline{\mathbf{a}}, b^i). \tag{5}$$

Next, we formulate the Bellman-type equation for the two Q functions, which we call the robust Harsanyi-Bellman equation. This update differs from the conventional approach by considering the posterior belief over other agents and the worst-case adversary.

**Definition 3.1.** We define the robust Harsanyi-Bellman equation for Q function as:

$$Q_*^i(s, \mathbf{a}, b^i) = \max_{\pi(\cdot|H,b)} \min_{\hat{\pi}(\cdot|H,\theta)} \sum_{\theta \in \Theta} p(\theta|H^i) \sum_{s' \in \mathcal{S}} \sum_{\hat{\mathbf{a}} \in \mathcal{A}} \overline{\mathcal{P}}(s'|s, \mathbf{a}, \hat{\pi}, \theta) \Big[$$
$$R(s, \overline{\mathbf{a}}) + \gamma \sum_{\mathbf{a}' \in \mathcal{A}} \pi(\mathbf{a}'|H', b') Q_*^i(s', \mathbf{a}', b'^i) \Big], \tag{6}$$

$$Q_*^i(s, \overline{\mathbf{a}}, b^i) = \max_{\pi(\cdot|H,b)} \min_{\hat{\pi}(\cdot|H,\theta)} R(s, \overline{\mathbf{a}}) + \gamma \sum_{s' \in \mathcal{S}} \mathcal{P}(s'|s, \overline{\mathbf{a}}) \sum_{\theta \in \Theta} p(\theta|H'^i)$$
$$\sum_{\overline{\mathbf{a}}' \in \mathcal{A}} \overline{\pi}(\overline{\mathbf{a}}'|H', b', \theta) Q_*^i(s', \overline{\mathbf{a}}', b'^i). \tag{7}$$

This Q function can be estimated via Temporal Difference (TD) loss.

**Proposition 3.1** (Convergence). Assume the belief is updated via Bayes' rule, the space of state, actions and belief are finite, updating value functions by robust Harsanyi-Bellman equation converge to the optimal value $Q_*^i(s, \overline{\mathbf{a}}, b^i)$ and $Q_*^i(s, \mathbf{a}, b^i)$.

*Proof sketch.* The proof is done by combining the standard convergence proof of Q function with adversaries and Bayesian belief update, and showing our Q function forms a contraction mapping. Next, applying Banach's fixed point theorem completes the proof. See full proof in Appendix. A.4.

### 3.2 ACTOR-CRITIC ALGORITHM FOR *ex interim* ROBUST EQUILIBRIUM

Armed with the robust Harsanyi-Bellman equation, we propose an actor-critic algorithm to achieve our proposed *ex interim* equilibrium with almost sure convergence under certain assumptions. We first derive the policy gradient theorem for robust c-MARL. Assume policies of robust agents and adversaries are parameterized by $\pi_\phi := (\pi_{\phi^i}^i)_{i \in \mathcal{N}}$ and $\hat{\pi}_{\hat{\phi}} := (\hat{\pi}_{\hat{\phi}^i}^i)_{i \in \mathcal{N}}$ respectively, forming a mixed policy $\overline{\pi}_{\phi,\hat{\phi}} = (1 - \theta) \cdot \pi_\phi + \theta \cdot \hat{\pi}_{\hat{\phi}}$. Define the performance for a robust agent $i$ in the episodic case as $J^i(\phi) = \mathbb{E}_{s \sim \rho^\pi(s)}[V^i(s, b^i)]$ and (zero-sum) adversary as $J^i(\hat{\phi}) = \mathbb{E}_{s \sim \rho^\pi(s)}[-V^i(s, b^i)]$, where $\rho^\pi(s)$ is the state visitation frequency. The policy gradients for $\pi_\phi$ and $\hat{\pi}_{\hat{\phi}}$ are then defined as:

**Theorem 3.1.** The policy gradient theorem for robust agent and adversary $i$ is:

$$\nabla_{\phi^i} J^i(\phi^i) = \mathbb{E}_{s \sim \rho^{\overline{\pi}}(s), \overline{\mathbf{a}} \sim \overline{\pi}_{\phi,\hat{\phi}}(\overline{\mathbf{a}}|H,b,\theta)} \left[ (1 - \theta^i) \nabla \log \pi_{\phi^i}(a^i|H^i, b^i) Q^i(s, \overline{\mathbf{a}}, b^i) \right], \tag{8}$$

$$\nabla_{\hat{\phi}^i} J^i(\hat{\phi}^i) = \mathbb{E}_{s \sim \rho^{\overline{\pi}}(s), \overline{\mathbf{a}} \sim \overline{\pi}_{\phi,\hat{\phi}}(\overline{\mathbf{a}}|H,b,\theta)} \left[ -\theta^i \nabla \log \hat{\pi}_{\hat{\phi}^i}(\hat{a}^i|H^i, \theta) Q^i(s, \overline{\mathbf{a}}, b^i) \right]. \tag{9}$$

The policy gradient naturally depends on $Q^i(s, \overline{\mathbf{a}}, b^i)$, which is related to policy gradient and decision of adversaries. Specifically, $\theta$ cuts off the gradient of robust agents $\pi_{\phi^i}$ with $\theta^i = 1$ and cut off the gradient of adversary $\hat{\pi}_{\hat{\phi}^i}$ with $\theta^i = 0$. The detailed derivation is deferred to Appendix. A.5.

**Convergence.** In zero-sum Markov games, achieving convergence through policy gradients remains challenging, with current theoretical results being dependent on specific conditions (Daskalakis et al., 2020; Zhang et al., 2020b; Kalogiannis et al., 2022). In this paper, we prove that, under certain assumptions in stochastic approximation theory (Borkar, 1997; Borkar & Meyn, 2000; Borkar, 2009), applying a two-timescale update for both adversaries and defenders, as stated in Theorem 3.1, guarantees almost sure convergence (*i.e.*, converge with probability 1) to an *ex interim* RMPBE. A detailed proof is provided in Appendix A.6 as an application of stochastic approximation. With this two-timescale update, the adversary's policy is updated on a faster timescale and is essentially equilibrated, while the defender's policy is updated on a slower timescale and remains quasi-static. Despite these advances, establishing finite sample, global convergence guarantees without restrictive assumptions remains an open problem, warranting future research.

Finally, we suggest update rules for critic and belief networks. Assuming the critic $Q_\psi^i(s, \overline{\mathbf{a}}, b^i)$ is parameterized by $\psi$. As for belief $b^i$, the calculation of $b^i$ via Bayes' rule requires assess to the policy

of other agents, which is not possible during deployment. To remedy this, we approximate the belief $b^i = \max_\xi p_\xi(\theta|H^i)$ using a neural network parameterized by $\xi$. The objectives to update critic and belief network are:

$$\min_\psi \left( R(s, \overline{\mathbf{a}}) - \gamma Q_\psi^i(s', \overline{\mathbf{a}}', b'^i) + Q_\psi^i(s, \overline{\mathbf{a}}, b^i) \right)^2, \tag{10}$$

$$\min_\xi -\theta \log \left( p_\xi(\theta|H^i) \right) - (1-\theta) \log \left( 1 - p_\xi(\theta|H^i) \right), \tag{11}$$

with critic trained via TD loss and belief network trained by binary cross entropy loss. See the pseudo-code for our algorithm in Appendix. B.

## 4 EXPERIMENTS

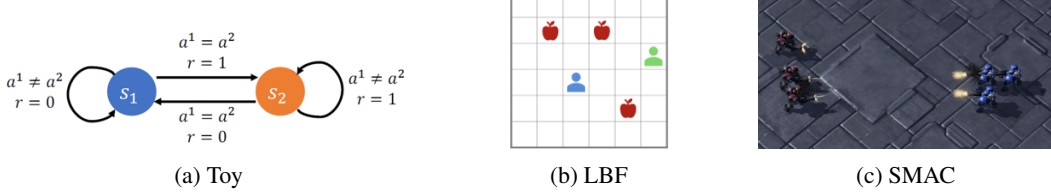

(a) Toy          (b) LBF          (c) SMAC

Figure 3: Environments used in our experiments. The toy iterative matrix game is proposed by Han et al. (2022). We use map *12x12-4p-3f-c* for LBF and map *4m vs 3m* for SMAC.

**Environments.** To validate the efficacy of our proposed approach, we conducted experiments on three benchmark cooperative MARL environments, as shown in Fig. 7. Environments include a toy iterative matrix game proposed by (Han et al., 2022), rewarding XNOR or XOR actions at different state, *12x12-4p-3f-c* of Level-Based Foraging (LBF) (Papoudakis et al., 2020) and *4m vs 3m* of the StarCraft Multi-agent Challenge (SMAC) (Samvelyan et al., 2019), which reduce to *3m* in the presence of an adversary.

**Baselines.** Our comparative study includes MADDPG (Lowe et al., 2017), M3DDPG (Li et al., 2019), MAPPO (Yu et al., 2021), RMAAC He et al. (2023), *ex ante* robust MAPPO (EAR-MAPPO), a MAPPO variant of (Phan et al., 2020; Zhang et al., 2020c) that considers *ex ante* equilibrium, which is also an ablation of our approach without belief. We dubbed our method *ex interim* robust MAPPO (EIR-MAPPO) and add an ideal case which grants access to true type, labelled "True Type". It's worth noting that we couldn't directly adapt M3DDPG onto the MAPPO framework due to its reliance on $Q(s, \mathbf{a})$, a component not compatible with MAPPO's use of $V(s)$ as a critic. More experiment details are given in Appendix. C. For fair comparison, all methods use the same codebase, network structure and hyperparameters. Code and demo videos available at https://github.com/DIG-Beihang/EIR-MAPPO.

**Evaluation protocol.** In each environment with $N$ cooperative agents, the robust policy was trained using five random seeds. Attack results were compiled by launching attacks on each of the $N$ agents using the same five seeds, yielding a total of $5 \times N$ attacks per environment. We plot all results with 95% confidence interval.

**Evaluated attacks.** We consider four types of threats. (1) Non-oblivious adversaries (Gleave et al., 2019): we fix the trained policy and deployed a zero-sum, worst-case adversarial policy to attack each agents separately. (2) Random agents: an agent perform random actions from a uniform distribution, possibly via hardware or software failure (labelled as "random"). (3) Noisy observations: we add $\ell_\infty$ bounded adversarial noise (Lin et al., 2020) with perturbation budgets $\epsilon \in \{0.2, 0.5, 1.0\}$ to the observation of an agent (denoted as "$\epsilon =$"). (4) Transferred adversaries: attackers initially train a policy on a surrogate algorithm, then directly transfer the attack to target other algorithms.

### 4.1 ROBUSTNESS OVER NON-OBLIVIOUS ATTACKS

We first evaluate our performance under the most arduous non-oblivious attack, where an adversary can manipulate any agent in cooperative tasks and execute arbitrary learned worst-case policy. The cooperation and attack performance on three environments are given in Fig. 4. Across all environments, our EIR-MAPPO consistently delivers robust performance close to the maximum

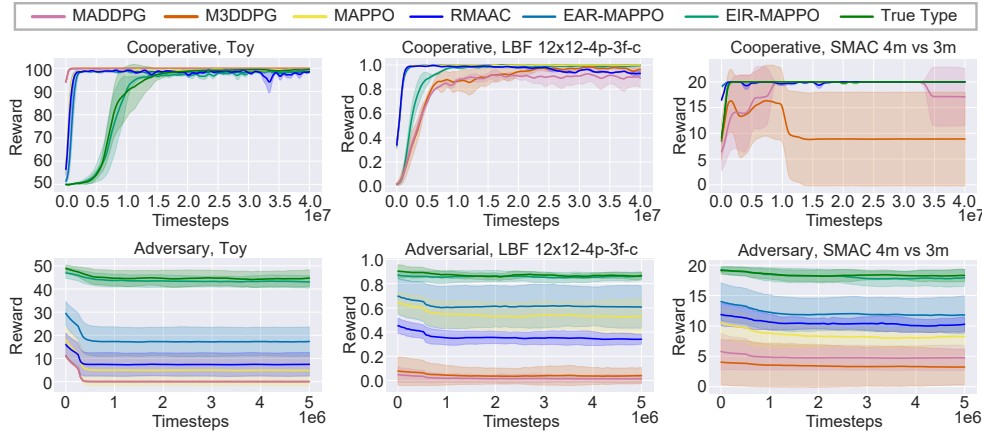

Figure 4: Cooperative and robust performance on three c-MARL environments. EIR-MAPPO achieves higher robust performance against non-oblivious adversaries and have cooperative performance on par with baselines. Reported on 5 seeds for cooperation and $5 \times N$ attacks.

reward achievable in each environment under attack (50 for Toy, 1.0 for LBF, 20 for SMAC), outperforming baselines by large margins and displays robustness equalling the ideal *True Type* defense. Concurrently, EIR-MAPPO maintains cooperative performance on par with MAPPO.

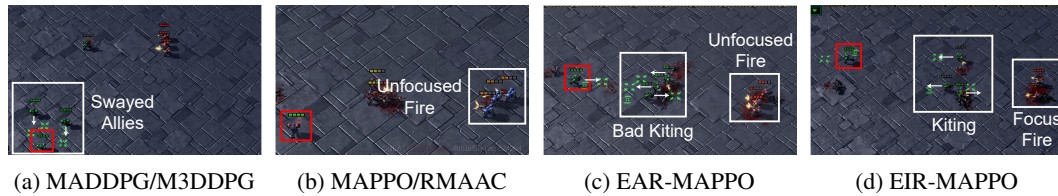

(a) MADDPG/M3DDPG    (b) MAPPO/RMAAC    (c) EAR-MAPPO    (d) EIR-MAPPO

Figure 5: Agent behaviors under attack. Red square indicates the adversary agent. Existing methods are either swayed, having unfocused fire or perform bad kiting. In contrast, our EIR-MAPPO learns kiting and focused fire simultaneously, under the presence of a worst-case adversary.

As illustrated in Fig. 5, a detailed examination of each method's behaviour under attack enriches our comprehension of robustness, with adversaries marked by a red square. First, MADDPG and M3DDPG can be easily swayed by adversaries. In Fig. 5a, a downward-moving adversary easily diverts two victims from the battle, resulting in a single victim facing three opponents. As for MAPPO and RMAAC, agents fail to master useful micromanagement strategies, such as kiting or focused fire during combat[1]. Consequently, the agent do not exihibit any cooperation skills under attack and are not skillful enough to win the game. As for EAR-MAPPO, agents occasionally demonstrate kiting but fall short in executing focused fire. They spread fire over two enemies instead of concentrating fire on one. Furthermore, even successful kiting can be compromised. In Fig. 5c, the adversary advances, causing two half-health agents to mistakenly believe that an ally is coming to its aid, and thus retreats to kite the enemy. This, however, leaves another low-health ally vulnerable to enemy fire and immediately being eliminated. Finally, we find that both EIR-MAPPO and *True Type* demonstrate focused fire and kiting, proving resistant to adversarial agents. Illustrated in Fig. 5d, two low-health agents retreat to avoid being eliminated, while an agent with high health advances to shield its allies, showcasing classic kiting behaviour. Moreover, allies coordinate to eliminate enemies, leaving one enemy nearly unscathed and another at half health.

## 4.2 ROBUSTNESS OVER VARIOUS TYPE OF ATTACKS

Apart from the worst-case oblivious adversary, c-MARL can encounter various uncertainties in real world, ranging from allies taking random actions, having uncertainties in observations, or a transferred

---

[1]Micromanagements are granular control strategies for agents to win in StarCraft II (Samvelyan et al., 2019). Kiting enables agents to evade enemy fire by stepping outside the enemy's attack range, thereby compelling the enemy to give chase rather than attack; focused fire requires taking enemies down one after another.

adversary trained on alternate algorithms. We examine the robust performance of all methods under such diverse uncertainties in Fig. 6. The rows signify uncertainties while the columns represent the evaluated methods. Diagonal entries (*i.e.*, blocks with the same uncertainty and evaluated method) denote non-oblivious attacks. For each uncertainty (column), the method of highest reward was marked by a red square. Furthermore, since the *True Type* represents an ideal scenario, if it secures the highest reward, we mark the method that gained the second-highest reward as well.

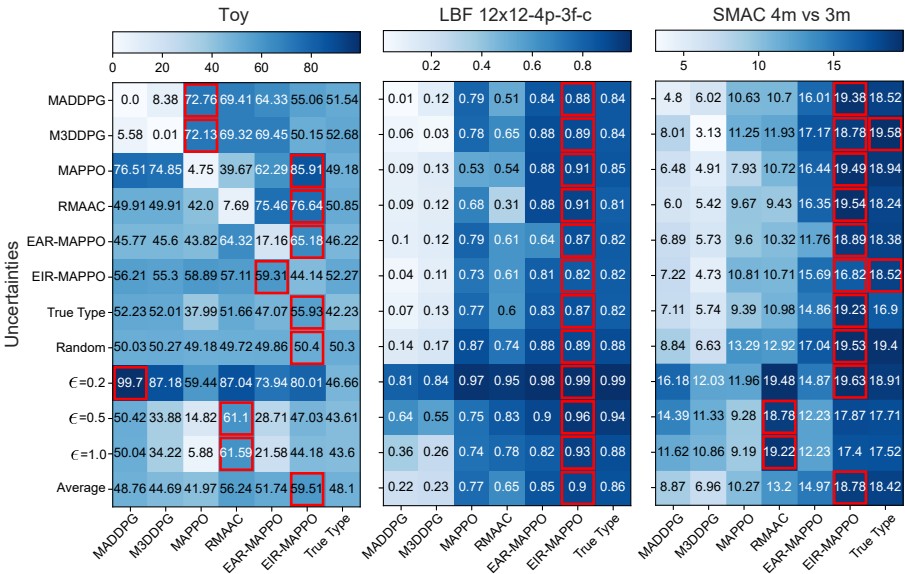

Figure 6: Evaluating robustness over diverse attacks, row indicates uncertainties and column indicates evaluated methods. EIR-MAPPO achieves improved robustness on almost all uncertainties in LBF and SMAC, while having higher robustness on average in Toy. Results averaged over $5 \times N$ attacks.

We first emphasize the effectiveness of our EIR-MAPPO approach. Considering the average reward gained under a broad range of uncertainties, our EIR-MAPPO surpasses baselines by 5.81% on Toy, 5.88% on LBF, and 25.45% on SMAC. Notably, EIR-MAPPO and True Type yields highest reward in almost all LBF and SMAC environments. In toy environment, owing to the existence of two pure-strategy equilibria, algorithms and attacks deploying deterministic strategies can occasionally yield higher rewards by chance. However, given its superior worst-case robustness, our EIR-MAPPO consistently delivers commendable results under all uncertainties.

Our second observation focuses on the relationship between action and observation perturbations. As an algorithm designed to counteract observation uncertainties, RMAAC is robust against observation perturbations, but fails to counter unseen action perturbations. In contrast, our EIR-MAPPO maintains its robustness against observation perturbations, even though it has not encountered the attack before. This resilience is due to the fact that observational attacks ultimately affect agents' choices of *actions*, which reduces observation uncertainty to a form of action uncertainty.

## 5 CONCLUSIONS

In this paper, we study robust c-MARL against Byzantine threat, where agents in this system to fail, or being compromised by an adversary. We frame the problem as a Dec-POMDP and define its solution as an *ex interim* RMPBE, such that the equilibrated policy weakly dominates *ex ante* solutions in prior research, when time goes to infinity. To actualize this equilibrium, we introduce Harsanyi-Bellman equation for value function updates, and an actor-critic algorithm with almost sure convergence under specific conditions. Experimental results show that under worst-case adversarial perturbation, our method can learn intricate and adaptive cooperation skills, and can withstand non-oblivious, random, observation-based, and transferred adversaries. As future work, we plan to apply our method to c-MARL applications, including robot swarm control (Hüttenrauch et al., 2019), traffic light management (Chu et al., 2019), and power grid maintenance (Xi et al., 2018).

## 6 ACKNOWLEDGEMENTS

This work was supported by the National Key Research and Development Plan of China (2022ZD0116405) and National Natural Science Foundation of China (62306025, 62022009, 62132010 and 62206009).

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

# APPENDIX FOR "BYZANTINE ROBUST COOPERATIVE MULTI-AGENT REINFORCEMENT LEARNING AS A BAYESIAN GAME"

## A PROOFS AND DERIVATIONS

### A.1 PROOF OF PROPOSITION 2.1

In this section, we proof a worst-case adversary always exist for any BARDec-POMDP with fixed defenders. Considering different numbers of adversaries (*i.e.*, $\sum_{i \in \mathcal{N}}$), with fixed defenders, the attackers can be seen as solving a POMDP or Dec-POMDP, where optimal solution exists.

First, if $\sum_{i \in \mathcal{N}} \theta_i = 0$, no adversary exists and the environment is reduced to a fully cooperative setting. Since adversary do not even exist, the problem becomes vacuous, and any $\hat{\pi}$ achieves the same (optimal) result.

Second, if $\sum_{i \in \mathcal{N}} \theta_i = 1$, the problem is reduced to a POMDP $\mathcal{G}^\alpha := \langle \mathcal{S}, \mathcal{O}, O, \mathcal{A}, \hat{\mathcal{P}}, \mathcal{P}^\alpha, R^\alpha, \gamma \rangle$ for adversary, where $\mathcal{S}$ the global state space, $\mathcal{O}$ the observation space for adversary, $O$ is the observation emission function, $\mathcal{A}$ is the action space of adversary, $R^\alpha : \mathcal{S} \times \mathcal{A} \to \mathbb{R}$ is the reward function for adversary. Actions taken by adversary $i$ are sampled via adversarial policy $\hat{a}_t^i \sim \hat{\pi}^i(\cdot|H_t^i, \theta)$. The environment transition for adversary is defined as $\hat{\mathcal{P}}(s_{t+1}|s_t, \hat{a}_t^i) = \mathcal{P}(s_{t+1}|s_t, \overline{\mathbf{a}}_t) \cdot \mathcal{P}^\alpha(\overline{\mathbf{a}}_t|\pi, \theta_t, \hat{a}_t^i)$, where $\overline{\mathbf{a}}_t$ is the mixed joint actions after perturbation in BARDec-POMDP. Here, $\mathcal{P}(s_{t+1}|s_t, \overline{\mathbf{a}}_t)$ is the environment transition in BARDec-POMDP, which combines with the transition $\mathcal{P}^\alpha(\overline{\mathbf{a}}_t|\pi, \theta, \hat{a}_t^i) = \prod_{i \in \mathcal{N}} \delta(\overline{a}_t^i - \hat{a}_t^i) \cdot \theta^i + \pi^i(\cdot|H_t^i) \cdot (1 - \theta^i)$ represents the decision of $\pi$, which is fixed and treated as a part of environment transition. Following the proof of Astrom et al. (1965), there always exists an optimal policy for POMDP. Thus, there exists an optimal adversary.

Third, with $\sum_{i \in \mathcal{N}} \theta_i > 1$, the problem is reduced to a Dec-POMDP $\langle \mathcal{N}, \mathcal{S}, \mathcal{O}^\alpha, O, \mathcal{A}, \hat{\mathcal{P}}, R^\alpha, \gamma \rangle$, with $\mathcal{S}$ the global state space, $\mathcal{O}^\alpha = \times_{i \in \{\theta^i = 1\}} \mathcal{O}^i$ is the observation space of adversaries, $O$ is the observation emission function, $\mathcal{A} = \times_{i \in \{\theta^i = 1\}} \mathcal{A}^i$ action space of adversaries, $R^\alpha : \mathcal{S} \times \mathcal{A}^\alpha \to \mathbb{R}$ is the reward function for adversary. Actions taken by adversary are sampled via adversarial policy $\hat{\mathbf{a}}_t \sim \hat{\pi}(\cdot|H_t, \theta)$. The environment transition *for adversary* is defined as $\hat{\mathcal{P}}(s_{t+1}|s_t, \hat{\mathbf{a}}_t) = \mathcal{P}(s_{t+1}|s_t, \overline{\mathbf{a}}_t) \cdot \mathcal{P}^\alpha(\overline{\mathbf{a}}_t|\pi, \theta, \hat{\mathbf{a}}_t)$. Here, $\mathcal{P}(s_{t+1}|s_t, \overline{\mathbf{a}}_t)$ is the environment transition for BARDec-POMDP, with the effect of the policy of $\pi$ merged in transition $\mathcal{P}^\alpha(\overline{\mathbf{a}}_t|\pi, \theta, \hat{\mathbf{a}}_t) = \prod_{i \in \mathcal{N}} \delta(\overline{a}_t^i - \hat{a}_t^i) \cdot \theta^i + \pi^i(\cdot|H_t^i) \cdot (1 - \theta^i)$, which is fixed and treated as a part of transition function. Following the proof of Bernstein et al. (2002), there always exists an optimal policy for Dec-POMDP. Thus, there exists an optimal adversary. $\square$

### A.2 PROOF OF PROPOSITION 2.2

We proof this by showing our BARDec-POMDP satisfies the requirement of Kakutani's fixed point theorem. Our proof is an extension of Kardeş et al. (2011), which shows equilibrium exists in robust stochastic games. In the following proof, we will first state existing results, then shows the existence of our *ex interim* RMPBE as a generalization of the proof of (Kardeş et al., 2011) considering Bayesian update and transforming action uncertainty as a kind of environment uncertainty. The existence proof of *ex ante* RMPBE rise as a corollary.

#### A.2.1 PRELIMINARIES

We first introduce existing results.

**Theorem A.1** (Kakutani's fixed point theorem). If $X$ is a closed, bounded, and convex set in a Euclidean space, and $\phi$ is an upper semicontinuous correspondence mapping $X$ into the family of closed, convex subsets of $X$, then $\exists x \in X, s.t. \ x \in \phi(x)$.

Next, we brief the definition of robust stochastic game and its equilibrium concept before we introduce the main result of Kardeş et al. (2011).

In a stochastic game $G = \langle \mathcal{N}, \mathcal{S}, \mathcal{A}, \mathcal{P}, R, \gamma \rangle$ with finite state and actions, where $\mathcal{N}$ is the indicator of agents, $\mathcal{S}$ is the state space, $\mathcal{A} = \times_{i \in \mathcal{N}} \mathcal{A}^i$ is the joint action space, $\mathcal{P} : \mathcal{S} \times \mathcal{A} \to \Delta(\mathcal{S})$ is the state transition probability, $R : \mathcal{S} \times \mathcal{A} \times \mathcal{N} \to \mathbb{R}^N$ is the reward function that maps state and actions to reward of each agent, $\gamma$ is the discount factor. Let $s \in \mathbf{S}$, $a^i \in \mathcal{A}^i$, with $\mathbf{a}$ the joint actions, $\pi : \mathcal{S} \to \Delta(\mathcal{A})$ is the policy for agent $i$ with $\pi^i(a^i|s)$, $r^i = R(s, a, i)$ is the reward for agent $i$. A *robust stochastic game* is a stochastic game with perturbed reward $\hat{r}_t \in \mathcal{R}^\alpha$ and environment transition $\hat{\mathcal{P}}(s'|s, \mathbf{a}) \in \mathcal{P}^\alpha$, with $\mathcal{R}^\alpha$ and $\mathcal{P}^\alpha$ are the bounded perturbation range.

Define the expected value function $V^i_{\pi, \hat{r}, \hat{\mathcal{P}}}(s)$ under perturbed reward and environment transition recursively through the following Bellman equation:

$$V^i_{\pi, \hat{r}, \hat{\mathcal{P}}}(s) = \sum_{\mathbf{a} \in \mathcal{A}} \pi^i(a^i|s) \prod_{j \neq i} \pi^j(a^j|s)[\hat{r}^i + \sum_{s' \in \mathcal{S}} \hat{\mathcal{P}}(s'|s, \mathbf{a}) V^i_\pi(s')],$$

the worst-case value function $V^i_{\pi, \hat{r}_*, \hat{\mathcal{P}}_*}(s)$ is defined as $V^i_{\pi, \hat{r}, \hat{\mathcal{P}}}(s) = \min_{\hat{r}, \hat{\mathcal{P}}} V^i_{\pi, \hat{r}_*, \hat{\mathcal{P}}_*}(s)$. The equilibrium policy, if exists, is thus $\forall i \in \mathcal{N}, s \in \mathcal{S}, a \in \mathcal{A}$, $\pi^i_*(\cdot|s) \in \arg\max_{\pi^i(\cdot|s)} V^i_\pi(s)$, with corresponding value as $V^i_{\{\pi^i_*, \pi^{-i}\}, \hat{r}_*, \hat{\mathcal{P}}_*}(s) = \max_{\pi^i} V^i_{\pi, \hat{r}_*, \hat{\mathcal{P}}_*}(s)$.

Next, let $\pi(\cdot|s) = \prod_{i \in \mathcal{N}} \pi^i(\cdot|s)$ be the joint policy, and the set of value functions for each agent as $V_{\pi, \hat{r}, \hat{\mathcal{P}}}(s) = (V^1_{\pi, \hat{r}, \hat{\mathcal{P}}}(s), V^2_{\pi, \hat{r}, \hat{\mathcal{P}}}(s), ..., V^N_{\pi, \hat{r}, \hat{\mathcal{P}}}(s))$. Define mappings $\beta$ and $\phi$ by $\beta(\pi^{-i}) = \{V^i | V^i = \max_{\pi^i} \min_{\hat{r}, \hat{\mathcal{P}}} V^i_{\{\pi^i, \pi^{-i}\}, \hat{r}, \hat{\mathcal{P}}}(s)\}$ and $\phi(\pi^{-i}) = \{\pi^i | \beta(\pi^{-i}) = V^i_{\{\pi^i_*, \pi^{-i}\}, \hat{r}_*, \hat{\mathcal{P}}_*}(s)\}$ to be the set of value functions and optimal policies.

**Theorem A.2** (Kardeş et al. (2011)). Assume uncertainties in transition probabilities and payoffs belongs to compact sets, all agents use stationary strategies, then the set of functions $\{V_{\pi, \hat{r}, \hat{\mathcal{P}}}(s), \hat{r} \in \mathcal{R}^\alpha, \hat{\mathcal{P}} \in \mathcal{P}^\alpha\}$ is equicontinuous and $V_{\pi, \hat{r}_*, \hat{\mathcal{P}}_*}(s)$ continuous on all its variables. $\phi(\pi^{-i})$ is an upper semicontinuous correspondence mapping $\Delta(\mathcal{A})$ in a convex and closed subsets of $\Delta(\mathcal{A})$, which satisfies the assumptions of Kakutani's fixed point theorem.

We are now ready to construct our proof.

### A.2.2 EXISTENCE OF EX INTERIM RMPBE

We first state the existence of ex interim RMPBE. The proof is conducted by transforming BARDec-POMDP to a Dec-POMDP with environmental uncertainties, and applying Bayesian update to value functions. Note that we include some results of Kardeş et al. (2011) for a self-contained proof.

First, by definition of our BARDec-POMDP, it can be transformed to a robust Dec-POMDP with adversary in environmental dynamics. This is done by combining environment transition $\mathcal{P}(s_{t+1}|s_t, \overline{\mathbf{a}})$ with action perturbation probability $\mathcal{P}^\alpha(\hat{\mathbf{a}}|\mathbf{a}, \hat{\pi}, \theta)$, resulting in $\overline{\mathcal{P}}(s_{t+1}|s_t, \mathbf{a}, \hat{\pi}, \theta) = \mathcal{P}(s_{t+1}|s_t, \hat{\mathbf{a}}) \cdot \mathcal{P}^\alpha(\hat{\mathbf{a}}|\mathbf{a}, \hat{\pi}, \theta)$. Thus, the robust Dec-POMDP can be seen as a particular case of stochastic game with shared reward, partial observations and uncertainties in perturbations. Thus, some results of Kardeş et al. (2011) can be taken for our proof. Before that, let us redefine some notations for clarity of our proof.

Let us redefine the expected value function $V_\theta(s)$ using the following Bellman equation:

$$V^i_{\pi, \hat{\pi}, \theta}(s) = \sum_{\mathbf{a} \in \mathcal{A}} \pi^i(a^i|H^i, b^i) \prod_{j \neq i} \pi^j(a^j|H^j, b^j)[r^i + \sum_{s' \in \mathcal{S}} \sum_{\hat{a} \in \mathcal{A}} \overline{\mathcal{P}}(s'|s, \mathbf{a}, \hat{\pi}, \theta) V_{\pi, \hat{\pi}, \theta}(s')].$$

Note that $V^i_{\pi, \hat{\pi}, \theta}(s)$ assumes the type $\theta$ is known, thus there is no uncertainty over $\theta$ and the function is not updated by robust Harsanyi-Bellman equation. The expected value function with belief, $V^i_{\pi, \hat{\pi}, b^i}(s)$, is defined by $V^i_{\pi, \hat{\pi}, b^i}(s) = \mathbb{E}_{p(\theta|H)}[V^i_{\pi, \hat{\pi}, \theta}(s)]$. The worst-case value function $V^i_{\pi, \hat{\pi}_*, \theta}(s)$ and $V^i_{\pi, \hat{\pi}_*, b^i}(s)$ are defined as $V^i_{\pi, \hat{\pi}_*, \theta}(s) = \min_{\hat{\pi}} V^i_{\pi, \hat{\pi}, \theta}(s)$ and $V^i_{\pi, \hat{\pi}_*, b^i}(s) = \min_{\hat{\pi}} V^i_{\pi, \hat{\pi}, b^i}(s)$. With optimal (equilibrium) policy defined as $\pi_*$, the corresponding value is $V^i_{\{\pi^i_*, \pi^{-i}\}, \hat{\pi}_*, b^i}(s) = \max_{\pi^i} V^i_{\pi, \hat{\pi}_*, b^i}(s)$. The mappings $\beta$ and $\phi$ are similarly defined $\beta(\pi^{-i}) = \{V^i | V^i = \max_{\pi^i} \min_{\hat{\pi}} V^i_{\{\pi^i, \pi^{-i}\}, \hat{\pi}, b^i}(s)\}$ and $\phi(\pi^{-i}) = \{\pi^i | \beta(\pi^{-i}) = V^i_{\{\pi^i_*, \pi^{-i}\}, \hat{\pi}, b^i}(s)\}$ to be the set of value functions and optimal policies. By Proposition. 3.1, the optimal value at each $s \in \mathcal{S}$ is unique, which we denote it as $v^i_s$.

**Lemma A.1** (Equicontinuity of $\{V^i_{\pi,\hat{\pi},b^i}(s), \hat{\pi} \in \Delta(\mathcal{A})\}$). For every $\epsilon > 0$, $\exists \delta > 0$, for any $(\pi_1, V_{\pi_1,\hat{\pi},b^i_1}(s'_1), b^i_2)$ and $(\pi_2, V_{\pi_2,\hat{\pi},b^i_2}(s'_2), b^i_2)$, $|\pi_1 - \pi_2| + |V_{\pi_1,\hat{\pi},b^i}(s'_1) - V_{\pi_2,\hat{\pi},b^i}(s'_2)| + |b_1 - b_2| < \delta$, then, $\forall \hat{\pi} \in \Delta(A), |V_{\pi_1,\hat{\pi},b^i}(s_1) - V_{\pi_2,\hat{\pi},b^i}(s_2)| < \epsilon$.

*Proof.* By Lemma 1 of Kardeş et al. (2011), the equicontinuity of $\{V^i_{\pi,\hat{\pi},\theta}(s), \hat{\pi} \in \Delta(\mathcal{A})\}$ holds, if we consider $\hat{\pi}$ as uncertainties in $\overline{\mathcal{P}}(s'|s, \mathbf{a}, \hat{\pi}, \theta)$. All we need now is to extend the proof considering belief $b^i$.

$$|V_{\pi_1,\hat{\pi},b^i}(s_1) - V_{\pi_2,\hat{\pi},b^i}(s_2)|$$

$$= |\mathbb{E}_{p(\theta|H^i_1)}[V_{\pi_1,\hat{\pi},\theta}(s_1)] - \mathbb{E}_{p(\theta|H^i_2)}[V_{\pi_2,\hat{\pi},\theta}(s_2)]|$$

$$= \left| \sum_{\theta \in \Theta} [p(\theta|H^i_1) \cdot V_{\pi_1,\hat{\pi},\theta}(s_1) - p(\theta|H^i_2) \cdot V_{\pi_2,\hat{\pi},\theta}(s_2)] \right|$$

$$= \left| \sum_{\theta \in \Theta} [p(\theta|H^i_1) \cdot (V_{\pi_1,\hat{\pi},\theta}(s_1) - V_{\pi_2,\hat{\pi},\theta}(s_2)) + (p(\theta|H^i_1) - p(\theta|H^i_2)) \cdot V_{\pi_2,\hat{\pi},\theta}(s_2)] \right|$$

$$\leq \sum_{\theta \in \Theta} [|p(\theta|H^i_1) \cdot (V_{\pi_1,\hat{\pi},\theta}(s_1) - V_{\pi_2,\hat{\pi},\theta}(s_2))| + |(p(\theta|H^i_1) - p(\theta|H^i_2)) \cdot V_{\pi_2,\hat{\pi},\theta}(s_2)|]$$

$$\leq \sum_{\theta \in \Theta} [|p(\theta|H^i_1)| \cdot |V_{\pi_1,\hat{\pi},\theta}(s_1) - V_{\pi_2,\hat{\pi},\theta}(s_2)| + |(p(\theta|H^i_1) - p(\theta|H^i_2)| \cdot |V_{\pi_2,\hat{\pi},\theta}(s_2)|]$$

Since $p(\theta|H^i_1)$ is a probability function, we have $p(\theta|H^i_1) \leq 1$. Since reward is finite, and $|V_{\pi_i,\hat{\pi},\theta}(s_i)|, i \in 1, 2$ is defined by discount factor $\gamma$, $|V_{\pi_i,\hat{\pi},\theta}(s_i)|$ is also bounded, *i.e.*, $|V_{\pi_i,\hat{\pi},\theta}(s_i)| \leq K$.

Now, let

$$|V_{\pi_1,\hat{\pi},\theta}(s_1) - V_{\pi_2,\hat{\pi},\theta}(s_2)| < \delta_1 = \frac{\min\{\epsilon, 1\}}{2 \cdot |\Theta|},$$

$$|(p(\theta|H^i_1) - p(\theta|H^i_2)| < \delta_2 = \frac{\min\{\epsilon, 1\}}{2 \cdot K},$$

and let $\delta = \min\{\delta_1, \delta_2\}$, we have:

$$|V_{\pi_1,\hat{\pi},b^i}(s_1) - V_{\pi_2,\hat{\pi},b^i}(s_2)|$$

$$\leq \sum_{\theta \in \Theta} [|p(\theta|H^i_1)| \cdot |V_{\pi_1,\hat{\pi},\theta}(s_1) - V_{\pi_2,\hat{\pi},\theta}(s_2)| + |(p(\theta|H^i_1) - p(\theta|H^i_2)| \cdot |V_{\pi_2,\hat{\pi},\theta}(s_2)|]$$

$$= \sum_{\theta \in \Theta} [|p(\theta|H^i_1)| \cdot \delta_1] + \sum_{\theta \in \Theta} [\delta_2 \cdot |V_{\pi_2,\hat{\pi},\theta}(s_2)|]$$

$$\leq \epsilon/2 + \epsilon/2 = \epsilon.$$

Thus, the set of functions $\{V^i_{\pi,\hat{\pi},b^i}(s), \hat{\pi} \in \Delta(\mathcal{A})\}$ is equicontinuous. $\qquad\square$

**Lemma A.2.** $\phi(\pi^{-i})$ is a convex set.

*Proof.* The proof follows Theorem 4 of Kardeş et al. (2011). By Lemma 2 of Kardeş et al. (2011), the pointwise minimum of an equicontinuous set of function is continuous, $V^i_{\pi,\hat{\pi}_*,b^i}(s) = \min_{\hat{\pi}} V^i_{\pi,\hat{\pi},b^i}(s)$ is continuous on all its variables. Besides, $V^i_{\pi,\hat{\pi}_*,b^i}(s)$ is defined by a discounted factor and is bounded. Thus, the maxima of $V^i_{\pi,\hat{\pi}_*,b^i}(s)$ exists.

Second, in Proposition 3.1, we have proof that updating Q function by robust Harsanyi-Bellman equation yields an optimal robust Q value. It rise as a simple corollary that the optimal V value, $V^i_{\{\pi^i_*,\pi^{-i}\},\hat{\pi}_*,b^i}(s)$ exists. By equality $V^i_{\{\pi^i_*,\pi^{-i}\},\hat{\pi}_*,b^i}(s) = \max_{\pi^i(a^i|H^i,b^i)} \min_{\hat{\pi}} \sum_{\mathbf{a} \in \mathcal{A}} \pi^i(a^i|H^i,b^i) \prod_{j \neq i} \pi^j(a^j|H^j,b^j) [r^i + \sum_{s' \in \mathcal{S}} \sum_{\hat{a} \in \mathcal{A}} \overline{\mathcal{P}}(s'|s, \mathbf{a}, \hat{\pi}, \theta) V^i_{\{\pi^i_*,\pi^{-i}\},\hat{\pi}_*,b^i}(s')$, $\phi(\pi^{-i}) \neq \varnothing$.

Third, by Lemma 3 of Kardeş et al. (2011), the value function considering worst-case adversary $V^i_{\pi,\hat{\pi}_*,b^i}(s) = \min_{\hat{\pi}} \sum_{\mathbf{a}\in\mathcal{A}} \pi^i(a^i|H^i,b^j) \prod_{j\neq i} \pi^j(a^j|H^j,b^j)[r^i + \sum_{s'\in\mathcal{S}} \sum_{\hat{a}\in\mathcal{A}} \sum_{\theta\in\Theta} p(\theta|H^i)\overline{\mathcal{P}}(s'|s,\mathbf{a},\hat{\pi},\theta)V_{\pi,\hat{\pi},b^i}(s')]$ is concave in $\pi^i$ with fixed $\pi^{-i}$ and $V^i_{\pi,\hat{\pi}_*,b^i}(s')$ [2].

Finally, we show the convexity of $\phi(\pi^{-i})$. Let $\pi^i_{1,*}, \pi^i_{2,*} \in \phi(\pi^{-i})$, with fixed $\pi^{-i}$. By definition of $V^i_{\{\pi^i_*,\pi^{-i}\},\hat{\pi}_*,b^i}(s)$, $\forall \pi, s \in \mathcal{S}, b \in \Delta(\Theta), i \in \mathcal{N}$, $V^i_{\{\pi^i_{1,*},\pi^{-i}\},\hat{\pi}_*,b^i}(s) = V^i_{\{\pi^i_{2,*},\pi^{-i}\},\hat{\pi}_*,b^i}(s) \geq V^i_{\pi,\hat{\pi}_*,b^i}(s)$. Thus, $\forall \lambda \in [0,1]$, we also have $\lambda V^i_{\{\pi^i_{1,*},\pi^{-i}\},\hat{\pi}_*,b^i}(s) + (1-\lambda)V^i_{\{\pi_{2,*},\pi^{-i}\},\hat{\pi}_*,b^i}(s) \geq V^i_{\pi,\hat{\pi}_*,b^i}(s)$. By concavity of $V^i_{\pi,\hat{\pi}_*,b^i}(s)$, $V^i_{\{\pi^i_*,\pi^{-i}\},\hat{\pi}_*,b^i}(s) = \lambda V^i_{\{\pi^i_{1,*},\pi^{-i}\},\hat{\pi}_*,b^i}(s) + (1-\lambda)V^i_{\{\pi^i_{2,*},\pi^{-i}\},\hat{\pi}_*,b^i}(s) \leq V^i_{\{\lambda\pi^i_{1,*}+(1-\lambda)\pi^i_{2,*},\pi^{-i}\},\hat{\pi}_*,b^i}(s)$. Since by Proposition 3.1, $V^i_{\pi,\hat{\pi}_*,b^i}(s)$ is optimal and unique. Thus, $\lambda\pi^i_{1,*} + (1-\lambda)\pi^i_{2,*} \in \phi(\pi^{-i})$, $\phi(\pi^{-i})$ is a convex set. $\qquad\square$

Next, we first introduce several lemmas, then show $\phi(x)$ is an upper semicontinuous correspondence.

**Lemma A.3.** Let $\mathcal{T}$ be the robust Harsanyi-Bellman operator defined in Appendix. A.6. $\mathcal{T}V^i_{\{\pi^i,\pi^{-i}\},\hat{\pi}_*,b^i}(s)$ is continuous in $\pi^{-i}$. The set $\{\mathcal{T}V_{\{\pi^i_*,\pi^{-i}\},\hat{\pi},b^i}(s)| \; V^i_{\{\pi^i_*,\pi^{-i}\},\hat{\pi}_*,b^i}(s)$ is bounded$\}$ is equicontinuous.

*Proof.* By Lemma 4 of Kardeş et al. (2011), $\mathcal{T}V^i_{\{\pi^i\pi^{-i}\},\hat{\pi}_*,\theta}(s)$ is continuous and the set $\{\mathcal{T}V_{\{\pi^i_*,\pi^{-i}\},\hat{\pi},b^i}(s)|V^i_{\{\pi^i_*,\pi^{-i}\},\hat{\pi}_*,\theta}(s)$ is bounded$\}$ is equicontinuous. Since $\mathcal{T}V^i_{\{\pi^i_*,\pi^{-i}\},\hat{\pi}_*,b^i}(s) = \mathcal{T}\mathbb{E}_{p(\theta|H)}[V^i_{\{\pi^i_*,\pi^{-i}\},\hat{\pi}_*,\theta}(s)]$, the expectation of a continuous function is still continuous, and the expectation over a equicontinuous set is still equicontinuous. This completes the proof. $\qquad\square$

**Lemma A.4.** Define the optimal value as $v^i = \{v^i_1, v^i_2, ...v^i_S\}$. If $\pi^i_n \to \pi^i$, $\pi^{-i}_n \to \pi^{-i}$, $\beta(\pi^{-i}_n) \to v^i$ and $\pi^i_n \in \phi(\pi^{-i}_n)$, then $\pi^i \in \phi(\pi^{-i})$, *i.e.*, $\phi(\pi^{-i})$ is an upper semicontinuous correspondence.

*Proof.* The proof is by Lemma 5 of Fink (1964). We re-write it here using our notation to make the proof self-contained. Specifically, $\forall s \in \mathcal{S}, H \in (\mathcal{O} \times \mathcal{A})^*, b \in \Delta(\Theta)$. Define function $f(\cdot)$ as $f(V_{\pi,\hat{\pi}_*,b^i}(s)) = \min_{\hat{\pi}(\hat{\mathbf{a}}|H,\theta)} \sum_{\mathbf{a}\in\mathcal{A}} \pi^i(a^i|H^i,b^i) \prod_{j\neq i} \pi^j(a^j|H^j,b^j)[r^i + \sum_{s'\in\mathcal{S}} \sum_{\hat{a}\in\mathcal{A}} \sum_{\theta\in\Theta} p(\theta|H^i)\overline{\mathcal{P}}(s'|s,\mathbf{a},\hat{\pi},b^i)V_{\pi,\hat{\pi}_*,b^i}(s')]$. Recall $v^i_s$ is the fixed point. Let $\pi^i_* \in \phi(\pi^{-i})$, $|f(v^i_s) - v^i_s| \leq |f(v^i_s) - f(\beta(\pi^{-i}_n|s))| + |f(\beta(\pi^{-i}_n|s)) - v^i_s| = |f(v^i_s) - f(\beta(\pi^{-i}_n|s))| + |\beta(\pi^{-i}_n|s) - v^i_s| \to 0$ as $n \to \infty$.

Now we need to show when $\pi^{-i}_n \to \pi^{-i}$ and $\beta(\pi^{-i}_n) \to v^i$, $\beta(\pi^{-i}|s) = v^i_s$. We have $|v^i_s - \mathcal{T}v^i_s| \leq |v^i_s - \beta(\pi^{-i}_n|s)| + |\beta(\pi^{-i}_n|s) - \mathcal{T}\beta(\pi^{-i}_n|s)| + |\mathcal{T}\beta(\pi^{-i}_n|s) - \mathcal{T}v^i_s|$. By our assumption, $\beta(\pi^{-i}_n) \to v^i$ and $\pi^i_n \in \phi(\pi^{-i}_n)$ as $n \to \infty$, $|v^i_s - \beta(\pi^{-i}_n|s)| \to 0$, $|\mathcal{T}\beta(\pi^{-i}_n|s) - \mathcal{T}v^i_s| \to 0$. By Lemma A.3, $|\beta(\pi^{-i}_n|s) - \mathcal{T}\beta(\pi^{-i}_n|s)| \to 0$. Thus, $|v^i_s - \mathcal{T}v^i_s| \to 0$ as $n \to \infty$. As $\beta(\pi^{-i}) = \{V^i|V^i = \max_{\pi^i} \min_{\hat{\pi}} V^i_{\{\pi^i,\pi^{-i}\},\hat{\pi},b^i}(s) = v^i_s\}$, $\beta(\pi^{-i}|s) = v^i_s$.

As we have $\beta(\pi^{-i}|s) = v^i_s$, we have $v^i_s = \mathcal{T}v^i_s$ and is a fixed point. Thus, $v^i_s = f(v^i_s) = \beta(\pi^{-i}|s) = \min_{\pi^i} V^i_{\{\pi^i,\pi^{-i}\},\hat{\pi}_*,b^i}(s)$. As a consequence, $\pi^i \in \phi(\pi^{-i})$, $\phi(\pi^{-i})$ is an upper semicontinuous correspondence. $\qquad\square$

Now, we have proofed $\phi$ is an upper semicontinuous correspondence (Lemma. A.4), mapping $\Delta(\mathcal{A})$ into the family of convex subsets of $\Delta(\mathcal{A})$ (Lemma. A.2). Since $\phi(\pi^{-i})$ is an upper semicontinuous correspondence, it is also a closed set for any $\pi$. Thus, the result satisfies the requirement of Kakutani's fixed point theorem, with equilibrium policy $\phi(\pi^{-i})$. $\qquad\square$

### A.2.3 EXISTENCE OF EX INTERIM RMPBE

The existence of *ex ante* RMPBE follows the proof of *ex interim* RMPBE, but having a prior belief $p(\theta)$ that is never updated. Since the expectation over $p(\theta)$ is a linear combination and $p(\theta)$ is bounded, the addition of $p(\theta)$ do not violate the convergence and continuity of value functions. The proof thus follows the result of *ex interim* RMPBE. $\qquad\square$

---

[2] Note that Lemma 3 of Kardeş et al. (2011) considers *cost*, which is the negative of *reward*. While in their proof, the function considering cost is convex. Taking the negative of a convex function is thus concave.

## A.3 Proof of Proposition 3.2

For convenience of notations, Let us redefine the value function under worst-case adversary and type $\theta$ at time $t$, but additionally adding $\pi, \hat{\pi}$ into the notation of $V_\theta(s)$ for clarity, resulting in

$$V_\theta^{\pi,\hat{\pi}*}(s) = \min_{\hat{\pi}(\cdot|H,\theta)} \sum_{\bar{\mathbf{a}} \in \mathcal{A}} \mathcal{P}^\alpha(\bar{\mathbf{a}}|\mathbf{a},\hat{\pi},\theta) \prod_{i \in \mathcal{N}} \pi^i(a^i|H^i)(R(s,\bar{\mathbf{a}}) + \gamma \sum_{s' \in \mathcal{S}} \mathcal{P}(s'|s,\bar{\mathbf{a}})V_\theta^{\pi,\hat{\pi}*}(s')).$$

We can then redefine the value function for *ex ante* RMPBE as $V_p^{\pi,\hat{\pi}*}(\theta)(s) = \mathbb{E}_{p(\theta)}[V_\theta^{\pi,\hat{\pi}*}(s)]$ and the value function for *ex interim* RMPBE as $V_b^{\pi,\hat{\pi}*}(s) = \mathbb{E}_{p(\theta|H)}[V_\theta^{\pi,\hat{\pi}*}(s)]$.

Next, for each $\theta \in \Theta$, with time $t \to \infty$, we have $b_t = p(\theta|H_t) \to \theta$ by consistency of Bayes' rule Diaconis & Freedman (1986), if $\forall \theta \in \Theta, p(\theta) \neq 0$ and with finite type space $\Theta$. The resulting value function at $t \to \infty$ is thus:

$$V_{b_t \to \infty}^{\pi_*^{EI},\hat{\pi}_*^{EI}}(s) = V_\theta^{\pi_*^{EI},\hat{\pi}_*^{EI}}(s) \geq V_\theta^{\pi,\hat{\pi}*}(s),$$

where $\hat{\pi}_*$ is always the optimal adversarial policy for current $\pi$. The rest follows. As for the *expected* return of *ex ante* robustness for $\theta$, we get:

$$V_{p(\theta)}^{\pi_*^{EA},\hat{\pi}_*^{EA}}(s) = \mathbb{E}_{p(\theta)} \left[ V_\theta^{\pi_*^{EA},\hat{\pi}_*^{EA}}(s) \right].$$

During evaluation, $\forall p(\theta)$, at $t \to \infty$, since current type $\theta$ is known and do not change, the return conditions on $\theta$, instead of expectations on $\theta$. As a consequence, the value function of *ex ante* and *ex interim* equilibrium is $V_\theta^{\pi_*^{EA},\hat{\pi}_*^{EA}}(s)$ and $V_\theta^{\pi_*^{EI},\hat{\pi}_*^{EI}}(s)$, respectively. At $t \to \infty$ and $\forall \theta \in \Theta$, we have $b_t = p(\theta|H_t) \to \theta$, and following inequality holds:

$$V_\theta^{\pi_*^{EI},\hat{\pi}_*^{EI}}(s) \geq V_\theta^{\pi_*^{EA},\hat{\pi}_*^{EA}}(s),$$

which use the fact $V_\theta^{\pi_*^{EI},\hat{\pi}_*^{EI}}(s) \geq V_\theta^{\pi,\hat{\pi}*}(s)$. Considering to the gap between $V_\theta^{\pi,\hat{\pi}*}(s)$ and $V_{p(\theta)}^{\pi,\hat{\pi}*}(s)$, a sufficient condition for this equality to hold is when the type space $\Theta$ contains one type only. Note that even at $t \to \infty$, the relation between *expected* value function $V_{p(\theta)}^{\pi_*^{EA},\hat{\pi}_{*,\theta}^{EA}}(s)$ of *ex ante* equilibrium and $V_\theta^{\pi_*^{EI},\hat{\pi}_*^{EI}}(s)$ of *ex interim* equilibrium is still not known. This is because the *ex ante* equilibrium can get high *expected* values by simply "believing" it in some prior $p(\theta)$ that yields high value. However, since the belief is not correct, the resulting policy is non-optimal in any type at $t \to \infty$. $\square$

## A.4 Proof of Proposition 3.3

*Overview.* We first proof the contraction mapping of $Q(s, \bar{\mathbf{a}}, b^i)$ by combining standard proof of the contraction mapping of Q function and Bayesian belief update. Afterwards, applying Banach's fixed point theorem completes the proof. The convergence of $Q(s, \mathbf{a}, b^i)$ follows the same vein.

We first show the proof for Q-function for $Q_*^i(s, \bar{\mathbf{a}}, b^i)$. The proof incorporates our robust Harsanyi-Bellman equation in the contraction mapping proof of robust MDP (Iyengar, 2005). Based on Bellman equation in Definition 3.1, let $\Delta(\Theta)$ be a probability over $\Theta$, the optimal Q-function of a contraction operator $\mathcal{T}$, defined from a generic function $Q : \mathcal{S} \times \mathcal{A} \times \Delta(\Theta) \to \mathbb{R}$ can be defined as:

$$(\mathcal{T}Q^i)(s, \bar{\mathbf{a}}, b^i) = \max_{\pi(\cdot|o,b)} \min_{\hat{\pi}(\cdot|o,\theta)} R(s,\bar{\mathbf{a}}) + \gamma \left[ \sum_{s' \in \mathcal{S}} \mathcal{P}(s'|s,\bar{\mathbf{a}}) \sum_{\theta \in \Theta} p(\theta|H^i) \right.$$
$$\left. \sum_{\hat{\mathbf{a}}' \in \mathcal{A}} \bar{\pi}(\bar{\mathbf{a}}'|H', b', \theta) Q_*^i(s', \bar{\mathbf{a}}', b'^i) \right].$$

Next, we show $\mathcal{T}$ forms a contraction operator, such that for any two Q function $Q_1^i$ and $Q_2^i$, assuming $\mathcal{T}Q_1^i(s, \bar{\mathbf{a}}, b^i) \geq \mathcal{T}Q_2^i(s, \bar{\mathbf{a}}, b^i)$, the following holds:

$$||\mathcal{T}Q_1^i - \mathcal{T}Q_2^i||_\infty \leq \gamma ||Q_1^i - Q_2^i||_\infty,$$

Specifically, for $\epsilon > 0$ and with fixed $a \in \mathcal{A}$, $s \in \mathcal{S}$, $b \in \Delta(\Theta)$, $H \in (\mathcal{O} \times \mathcal{A})^*$ for $Q_1$ and $Q_2$,

$$\min_{\hat{\pi}(\cdot|H,\theta)} R(s,\hat{\mathbf{a}}) + \gamma \left[ \sum_{s' \in \mathcal{S}} \mathcal{P}(s'|s,\overline{\mathbf{a}}) \sum_{\theta \in \Theta} p(\theta|H^i) \sum_{\overline{\mathbf{a}}' \in \mathcal{A}} \overline{\pi}(\hat{\mathbf{a}}'|H',b',\theta) Q_1^i(s',\overline{\mathbf{a}}',b'^i) \right] \geq \mathcal{T}Q_1^i - \epsilon.$$

Note that updating belief by Bayes' rule is required for consistency of $b'^i$. We require fixed $H^i$ as well since the calculation of belief and Q function depends on $H^i$. Now we can also choose a conditional policy measure of the adversary $\hat{\pi}_s$, such that:

$$\mathbb{E}_{\hat{\pi}_s} \left[ R(s,\hat{\mathbf{a}}) + \gamma \left[ \sum_{s' \in \mathcal{S}} \mathcal{P}(s'|s,\hat{\mathbf{a}}) \sum_{\theta \in \Theta} p(\theta|H^i) \sum_{\hat{\mathbf{a}}' \in \mathcal{A}} \overline{\pi}(\hat{\mathbf{a}}'|H',b',\theta) Q_2^i(s',\hat{\mathbf{a}}',b'^i) \right] \right] \leq$$

$$\min_{\hat{\pi}(\cdot|o,\theta)} R(s,\hat{\mathbf{a}}) + \gamma \left[ \sum_{s' \in \mathcal{S}} \mathcal{P}(s'|s,\hat{\mathbf{a}}) \sum_{\theta \in \Theta} p(\theta|H^i) \sum_{\hat{\mathbf{a}}' \in \mathcal{A}} \overline{\pi}(\hat{\mathbf{a}}'|H',b',\theta) Q_2^i(s',\hat{\mathbf{a}}',b'^i) \right] + \epsilon.$$

Then,

$$0 \leq \mathcal{T}Q_1^i - \mathcal{T}Q_2^i$$
$$\leq \left( \min_{\hat{\pi}(\cdot|H,\theta)} R(s,\hat{\mathbf{a}}) + \gamma \left[ \sum_{s' \in \mathcal{S}} \mathcal{P}(s'|s,\overline{\mathbf{a}}) \sum_{\theta \in \Theta} p(\theta|H^i) \sum_{\hat{\mathbf{a}}' \in \mathcal{A}} \overline{\pi}(\hat{\mathbf{a}}'|H',b',\theta) Q_1^i(s',\hat{\mathbf{a}}',b'^i) \right] + \epsilon \right) -$$
$$\left( \min_{\hat{\pi}(\cdot|H,\theta)} R(s,\overline{\mathbf{a}}) + \gamma \left[ \sum_{s' \in \mathcal{S}} \mathcal{P}(s'|s,\overline{\mathbf{a}}) \sum_{\theta \in \Theta} p(\theta|H^i) \sum_{\overline{\mathbf{a}}' \in \mathcal{A}} \overline{\pi}(\overline{\mathbf{a}}'|H',b',\theta) Q_2^i(s',\overline{\mathbf{a}}',b'^i) \right] \right)$$
$$\leq \left( \mathbb{E}_{\hat{\pi}_s} \left[ R(s,\overline{\mathbf{a}}) + \gamma \left[ \sum_{s' \in \mathcal{S}} \mathcal{P}(s'|s,\overline{\mathbf{a}}) \sum_{\theta \in \Theta} p(\theta|H^i) \sum_{\overline{\mathbf{a}}' \in \mathcal{A}} \overline{\pi}(\overline{\mathbf{a}}'|H',b',\theta) Q_1^i(s',\overline{\mathbf{a}}',b'^i) \right] \right] + \epsilon \right) -$$
$$\left( \mathbb{E}_{\hat{\pi}_s} \left[ R(s,\overline{\mathbf{a}}) + \gamma \left[ \sum_{s' \in \mathcal{S}} \mathcal{P}(s'|s,\overline{\mathbf{a}}) \sum_{\theta \in \Theta} p(\theta|H^i) \sum_{\overline{\mathbf{a}}' \in \mathcal{A}} \overline{\pi}(\overline{\mathbf{a}}'|H',b',\theta) Q_2^i(s',\overline{\mathbf{a}}',b'^i) \right] \right] - \epsilon \right)$$
$$= \gamma \mathbb{E}_{\hat{\pi}_s} \left[ Q_1^i - Q_2^i \right] + 2\epsilon \leq \gamma \mathbb{E}_{\hat{\pi}_s} \left| Q_1^i - Q_2^i \right| + 2\epsilon \leq \gamma \mathbb{E}_{\hat{\pi}_s} ||Q_1^i - Q_2^i||_\infty + 2\epsilon.$$

Thus, we have:

$$||\mathcal{T}Q_1^i - \mathcal{T}Q_2^i||_\infty \leq \gamma ||Q_1^i - Q_2^i||_\infty + 2\epsilon,$$

and since by definition, $\epsilon$ is arbitrary, then we have $||\mathcal{T}Q_1^i - \mathcal{T}Q_2^i||_\infty \leq \gamma ||Q_1^i - Q_2^i||_\infty$.

Finally, since $\mathcal{T}$ is a contraction operator on a Banach space, by Banach's fixed point theorem, updating $Q^i(s,\overline{\mathbf{a}},b^i)$ by Bellman operator $\mathcal{T}$ will converge to the optimal value function $Q_*^i(s,\overline{\mathbf{a}},b^i)$.

In the same way, the convergence of $Q^i(s,\mathbf{a},b^i)$ is again done by robust Harsanyi-Bellman equation, $Q_*^i(s,\mathbf{a},b^i) = \max_{\pi(\cdot|H,b)} \min_{\hat{\pi}(\cdot|H,\theta)} \sum_{\theta \in \Theta} p(\theta|H^i) \sum_{s' \in \mathcal{S}} \sum_{\hat{\mathbf{a}} \in \mathcal{A}} \overline{\mathcal{P}}(s'|s,\mathbf{a},\hat{\pi},\theta)[R(s,\overline{\mathbf{a}}) + \gamma \sum_{\mathbf{a}' \in \mathcal{A}} \pi(\mathbf{a}'|H',b') Q_*^i(s',\mathbf{a}',b'^i)]$. Expanding the function in the same way above completes the proof. $\qquad \square$

## A.5  PROOF OF THEOREM 4.1

We first discuss the policy gradient with $\pi_{\phi^i}(a^i|H^i, b^i)$:

$$
\begin{aligned}
\nabla_{\phi^i} V^i(s, b^i) =& \nabla_{\phi^i} \left[ \sum_{\overline{\mathbf{a}} \in \mathcal{A}} \overline{\pi}_{\phi, \hat{\phi}}(\overline{\mathbf{a}}|H, b, \theta) Q^i(s, \overline{\mathbf{a}}, b^i) \right] \\
=& \nabla_{\phi^i} \left[ \sum_{\overline{\mathbf{a}} \in \mathcal{A}} \left( (1-\theta) \cdot \pi_\phi(\mathbf{a}|H, b) + \theta \cdot \hat{\pi}_{\hat{\phi}}(\hat{\mathbf{a}}|H, \theta) \right) Q^i(s, \overline{\mathbf{a}}, b^i) \right] \\
=& \sum_{\overline{\mathbf{a}} \in \mathcal{A}} \left[ (1-\theta^i) \nabla_{\phi^i} \pi_{\phi^i}(a^i|H^i, b^i) \cdot Q^i(s, \overline{\mathbf{a}}, b^i) + \overline{\pi}_{\phi, \hat{\phi}}(\overline{\mathbf{a}}|H, b, \theta) \nabla_{\phi^i} Q^i(s, \overline{\mathbf{a}}, b^i) \right] \\
=& \sum_{\overline{\mathbf{a}} \in \mathcal{A}} \left[ (1-\theta^i) \nabla_{\phi^i} \pi_{\phi^i}(a^i|H^i, b^i) \cdot Q^i(s, \overline{\mathbf{a}}, b^i) + \overline{\pi}_{\phi, \hat{\phi}}(\hat{\mathbf{a}}|H, b, \theta) \nabla_{\phi^i} \Big[ R(s, \overline{\mathbf{a}}) \right. \\
& \left. + \gamma \sum_{s' \in \mathcal{S}} \sum_{\overline{\mathbf{a}}' \in \mathcal{A}} \mathcal{P}(s'|s, \overline{\mathbf{a}}) \sum_{\theta \in \Theta} p(\theta|H) \overline{\pi}_{\phi, \hat{\phi}}(\overline{\mathbf{a}}'|H', b', \theta) Q^i(s', \overline{\mathbf{a}}', b'^i) \Big] \right], \\
=& \sum_{\overline{\mathbf{a}} \in \mathcal{A}} \left[ (1-\theta^i) \nabla_{\phi^i} \pi_{\phi^i}(a^i|o^i, b^i) \cdot Q^i(s, \overline{\mathbf{a}}, b^i) + \overline{\pi}_{\phi, \hat{\phi}}(\overline{\mathbf{a}}|H, b, \theta) \Big[ \gamma (1-\theta^i) \nabla_{\phi^i} \right. \\
& \left. \pi_{\phi^i}(a'|H', b') + \gamma \sum_{s' \in \mathcal{S}} \sum_{\overline{\mathbf{a}}' \in \mathcal{A}} \mathcal{P}(s'|s, \overline{\mathbf{a}}) \overline{\pi}_{\phi, \hat{\phi}}(\overline{\mathbf{a}}'|H', b', \theta) \sum_{\theta \in \Theta} p(\theta|H) \nabla_{\phi^i} Q^i(s', \overline{\mathbf{a}}', b'^i) \Big] \right], \\
=& \sum_{s' \in \mathcal{S}} \sum_{t=0}^{\infty} Pr(s \to s', t, \overline{\pi}) \sum_{\hat{\mathbf{a}} \in \mathcal{A}} (1-\theta^i) \nabla_{\phi^i} \pi_{\phi^i}(\mathbf{a}^i|H^i, b^i) \cdot Q^i(s, \overline{\mathbf{a}}, b^i).
\end{aligned}
$$

Considering $\nabla_{\phi^i} J^i(\phi^i)$, we have

$$
\begin{aligned}
\nabla_{\phi^i} J^i(\phi^i) =& \nabla_{\phi^i} V^i(s_0, b^i) \\
=& \sum_{s \in \mathcal{S}} \sum_{t=0}^{\infty} Pr(s_0 \to s, t, \overline{\pi}) \sum_{\overline{\mathbf{a}} \in \mathcal{A}} (1-\theta^i) \nabla_{\phi^i} \pi_{\phi^i}(a^i|H^i, b^i) \cdot Q^i(s, \overline{\mathbf{a}}, b^i) \\
=& \sum_{s \in \mathcal{S}} \eta(s) \sum_{\overline{\mathbf{a}} \in \mathcal{A}} (1-\theta^i) \nabla_{\phi^i} \pi_{\phi^i}(a^i|H^i, b^i) \cdot Q^i(s, \overline{\mathbf{a}}, b^i) \\
=& \sum_{s' \in \mathcal{S}} \eta(s') \sum_{s \in \mathcal{S}} \frac{\eta(s)}{\sum_{s' \in \mathcal{S}} \eta(s')} \sum_{\overline{\mathbf{a}} \in \mathcal{A}} (1-\theta^i) \nabla_{\phi^i} \pi_{\phi^i}(a^i|H^i, b^i) \cdot Q^i(s, \overline{\mathbf{a}}, b^i) \\
=& \sum_{s' \in \mathcal{S}} \eta(s') \sum_{s \in \mathcal{S}} \rho^{\overline{\pi}}(s) \sum_{\overline{\mathbf{a}} \in \mathcal{A}} (1-\theta^i) \pi_{\phi^i}(a^i|H^i, b^i) \cdot Q^i(s, \overline{\mathbf{a}}, b^i) \\
\propto& \sum_{s \in \mathcal{S}} \rho^{\overline{\pi}}(s) \sum_{\overline{\mathbf{a}} \in \mathcal{A}} (1-\theta^i) \nabla_{\phi^i} \pi_{\phi^i}(a^i|H^i, b^i) \cdot Q^i(s, \overline{\mathbf{a}}, b^i).
\end{aligned}
$$

Using the log-derivative trick, we have:

$$
\begin{aligned}
\nabla_{\phi^i} J^i(\phi^i) \propto& \sum_{s \in \mathcal{S}} \rho^{\overline{\pi}}(s) \sum_{\overline{\mathbf{a}} \in \mathcal{A}} (1-\theta^i) \nabla_{\phi^i} \pi_{\phi^i}(a^i|H^i, b^i) \cdot Q^i(s, \overline{\mathbf{a}}, b^i) \\
=& \sum_{s \in \mathcal{S}} \rho^{\overline{\pi}}(s) \sum_{\overline{\mathbf{a}} \in \mathcal{A}} (1-\theta^i) \pi_{\phi^i}(a^i|H^i, b^i) \cdot Q^i(s, \overline{\mathbf{a}}, b^i) \frac{\nabla_{\phi^i} \pi_{\phi^i}(a^i|H^i, b^i)}{\pi_{\phi^i}(a^i|H^i, b^i)} \\
=& \mathbb{E}_{s \sim \rho^{\overline{\pi}}(s), \overline{\mathbf{a}} \sim \overline{\pi}(\cdot|H, b, \theta)} [(1-\theta^i) \nabla_{\phi^i} \ln \pi_{\phi^i}(a^i|H^i, b^i) \cdot Q^i(s, \overline{\mathbf{a}}, b^i)].
\end{aligned}
$$

The proof of $\pi_{\hat{\phi}^i}(\hat{a}^i | o^i, \theta)$ basically follows the same method.

$$
\begin{aligned}
\nabla_{\hat{\phi}^i} V^i(s, b^i) =& \nabla_{\hat{\phi}^i} \left[ \sum_{\overline{\mathbf{a}} \in \mathcal{A}} \overline{\pi}_{\phi, \hat{\phi}}(\overline{\mathbf{a}} | H, b, \theta) Q^i(s, \overline{\mathbf{a}}, b^i) \right] \\
=& \nabla_{\hat{\phi}^i} \left[ \sum_{\overline{\mathbf{a}} \in \mathcal{A}} \left( (1 - \theta) \cdot \pi_\phi(\mathbf{a} | H, b) + \theta \cdot \hat{\pi}_{\hat{\phi}}(\hat{\mathbf{a}} | H, \theta) \right) Q^i(s, \overline{\mathbf{a}}, b^i) \right] \\
=& \sum_{\overline{\mathbf{a}} \in \mathcal{A}} \left[ \theta^i \cdot \nabla_{\hat{\phi}^i} \hat{\pi}_{\hat{\phi}^i}(\hat{a}^i | H^i, \theta) \cdot Q^i(s, \overline{\mathbf{a}}, b^i) + \overline{\pi}_{\phi, \hat{\phi}}(\hat{\mathbf{a}} | H, b, \theta) \nabla_{\hat{\phi}^i} Q^i(s, \overline{\mathbf{a}}, b^i) \right] \\
=& \sum_{\overline{\mathbf{a}} \in \mathcal{A}} \left[ \theta^i \cdot \nabla_{\hat{\phi}^i} \hat{\pi}_{\hat{\phi}^i}(\hat{a}^i | H^i, \theta) \cdot Q^i(s, \overline{\mathbf{a}}, b^i) + \overline{\pi}_{\phi, \hat{\phi}}(\hat{\mathbf{a}} | H, b, \theta) \nabla_{\hat{\phi}^i} \left[ R(s, \overline{\mathbf{a}}) \right. \right. \\
& \left. \left. + \gamma \sum_{s' \in \mathcal{S}} \sum_{\overline{\mathbf{a}}' \in \mathcal{A}} \mathcal{P}(s' | s, \overline{\mathbf{a}}) \sum_{\theta \in \Theta} p(\theta | H) \overline{\pi}_{\phi, \hat{\phi}}(\overline{\mathbf{a}}' | H', b', \theta) Q^i(s', \hat{\mathbf{a}}', b'^i) \right] \right], \\
=& \sum_{\overline{\mathbf{a}} \in \mathcal{A}} \left[ \theta^i \cdot \nabla_{\hat{\phi}^i} \hat{\pi}_{\hat{\phi}^i}(\hat{a}^i | H^i, \theta) \cdot Q^i(s, \overline{\mathbf{a}}, b^i) + \overline{\pi}_{\phi, \hat{\phi}}(\overline{\mathbf{a}} | H, b, \theta) \left[ \gamma \cdot \theta^i \cdot \nabla_{\hat{\phi}^i} \right. \right. \\
& \left. \left. \hat{\pi}_{\hat{\phi}^i}(\hat{a}'^i | H'^i, \theta) + \gamma \sum_{s' \in \mathcal{S}} \sum_{\overline{\mathbf{a}}' \in \mathcal{A}} \mathcal{P}(s' | s, \overline{\mathbf{a}}) \sum_{\theta \in \Theta} p(\theta | H) \overline{\pi}_{\phi, \hat{\phi}}(\hat{\mathbf{a}}' | H', b', \theta) \nabla_{\phi^i} Q^i(s', \overline{\mathbf{a}}', b'^i) \right] \right], \\
=& \sum_{s' \in \mathcal{S}} \sum_{t=0}^{\infty} Pr(s \to s', t, \overline{\pi}) \sum_{\overline{\mathbf{a}} \in \mathcal{A}} \theta^i \cdot \nabla_{\hat{\phi}^i} \hat{\pi}_{\hat{\phi}^i}(\hat{a}^i | H^i, \theta) \cdot Q^i(s, \overline{\mathbf{a}}, b^i).
\end{aligned}
$$

Considering $\nabla_{\phi^i} J^i(\phi^i)$ and use the log-derivative trick same as above, we get:

$$
\begin{aligned}
\nabla_{\hat{\phi}^i} J^i(\hat{\phi}^i) \propto& \sum_{s \in \mathcal{S}} \rho^{\overline{\pi}}(s) \sum_{\overline{\mathbf{a}} \in \mathcal{A}} \theta^i \cdot \nabla_{\hat{\phi}^i} \hat{\pi}_{\hat{\phi}^i}(\hat{a}^i | H^i, \theta) \cdot Q^i(s, \overline{\mathbf{a}}, b^i) \\
=& \sum_{s \in \mathcal{S}} \rho^{\overline{\pi}}(s) \sum_{\overline{\mathbf{a}} \in \mathcal{A}} \theta^i \hat{\pi}_{\hat{\phi}}(\hat{a}^i | H^i, \theta) \cdot Q^i(s, \hat{\mathbf{a}}, b^i) \frac{\nabla_{\hat{\phi}^i} \hat{\pi}_{\hat{\phi}^i}(\hat{a}^i | H^i, \theta)}{\hat{\pi}_{\hat{\phi}^i}(\hat{a}^i | H^i, \theta)} \\
=& \mathbb{E}_{s \sim \rho^{\overline{\pi}}(s), \overline{\mathbf{a}} \sim \overline{\pi}(\cdot | H, b, \theta)} [\theta^i \nabla_{\hat{\phi}^i} \ln \hat{\pi}_{\hat{\phi}^i}(\hat{a}^i | H^i, \theta) \cdot Q^i(s, \overline{\mathbf{a}}, b^i)].
\end{aligned}
$$

This completes the proof. $\qquad \square$

## A.6 Convergence Proof of Theorem 3.1

We proof this via stochastic approximation theory of Borkar (Borkar, 1997; Borkar & Meyn, 2000; Borkar, 2009), where the robust agent is quasi-static and the adversary is essentially equilibrated (Borkar & Meyn, 2000). One notable difference is, since the type in our work was sampled from prior distribution $\theta \sim p(\theta)$, we take the expectation with respect to $p(\theta)$ to follow the notation of stochastic approximation theory.

**Proposition A.1** (Convergence). Theorem 3.1 in main paper converge to robust Bayesian Markov Perfect equilibrium *a.s.* if the following assumption holds.

**Assumption A.1.** Given step $n$, learning rate of $\pi$ and $\hat{\pi}$ as $\alpha(n)$ and $\beta(n)$ with $\alpha(n), \beta(n) \in (0, 1)$, denote the probability of having an adversary as $p^{\theta^i} = p(\theta^i = 1)$, such that $\forall i \in \mathcal{N}$, $\sum_t \alpha(n)(1 - p^{\theta^i}) = \sum_n \beta(n) p^{\theta^i} = \infty$, $\sum_t (\alpha(n)(1 - p^{\theta^i}))^2 + (\beta(n) p^{\theta^i})^2 < \infty$, $\frac{\alpha(n)(1 - p^{\theta^i})}{\beta(n) p^{\theta^i}} \to 0$.

*Note that the assumption slightly differs from standard stochastic approximation, since adversaries and robust agents are not uniformly explored.*

**Assumption A.2.** $\forall i \in \mathcal{N}, \theta \in \Theta$, $Q^i(s, \mathbf{a}, \theta)$ is Lipshitz continuous. As a corollary, $Q^i(s, \mathbf{a}, b^i) = \mathbb{E}_{p(\theta | H^i)}[Q^i(s, \mathbf{a}, \theta)]$ is Lipshitz continuous.

**Assumption A.3.** Let $\nu(s, a, \theta)$ denote the number of visit to state $s$ and action $a$ under $\theta^i$. $\forall s, a, \theta, \nu(s, a, \theta) \to \infty$.

**Assumption A.4.** The error in stochastic approximation (*i.e.*, inaccuracy in critic value, environment noise, belief*etc.*) constitutes martingale difference sequences with respect to the increasing $\sigma$-fields.

**Assumption A.5.** $\forall \theta \in \Theta$, a global asymptotically stable *ex interim* equilibrium $(\pi_*^{EI}, \hat{\pi}_*^{EI})$ exists.

**Assumption A.6.** $\forall \theta \in \Theta, \sup_t(||\pi_n^{EI}|| + ||\hat{\pi}_n^{EI}||) < \infty$.

*proof.* We can write the update rule of $\pi$ and $\hat{\pi}$ in their Ordinary Differential Equation (ODE) form:

$$
\begin{aligned}
\mathbb{E}_{\theta^i \sim p(\theta^i)}[\pi_{n+1}^i] =& \mathbb{E}_{\theta^i \sim p(\theta^i)}[\pi_n^i] + \alpha(\nu(s,a))(1 - p^{\theta^i})\mathbb{E}_{\theta^i \sim p(\theta^i)}[\nabla \log \pi_n^i (R(s, \overline{\mathbf{a}}) \\
& + \sum_{s' \in \mathcal{S}} \mathcal{P}(s'|s, \overline{\mathbf{a}}) \sum_{\theta \in \Theta} p(\theta|H'^i) \sum_{\overline{\mathbf{a}} \in \mathcal{A}} \overline{\pi}_n(\overline{\mathbf{a}}'|H', b', \theta) Q_n^i(s', \overline{\mathbf{a}}', b'^i))], \\
\mathbb{E}_{\theta^i \sim p(\theta^i)}[\hat{\pi}_{n+1}^i] =& \mathbb{E}_{\theta^i \sim p(\theta^i)}[\hat{\pi}_n^i] + \beta(\nu(s,a)) p^{\theta^i}\mathbb{E}_{\theta^i \sim p(\theta^i)}[\nabla \log \hat{\pi}_n^i (R(s, \overline{\mathbf{a}}) \\
& + \sum_{s' \in \mathcal{S}} \mathcal{P}(s'|s, \overline{\mathbf{a}}) \sum_{\theta \in \Theta} p(\theta|H'^i) \sum_{\hat{\mathbf{a}} \in \mathcal{A}} \overline{\pi}_n(\overline{\mathbf{a}}'|H', b', \theta) Q_n^i(s', \overline{\mathbf{a}}', b'^i))].
\end{aligned}
$$

where we assume $Q_n^i(s', \overline{\mathbf{a}}', b'^i)$ is the learned critic, updated at a faster timescale than $\alpha(n)$ and $\beta(n)$, following Assumption 1.1-1.5, such that $Q^i$ is essentially equilibrated. The error terms are embedded in $Q_n^i(s', \mathbf{a}', H'^i)$. Thus, the update rule of $\pi^i$ and $\hat{\pi}^i$ follows the general update rule of stochastic optimization (Borkar, 2009):

$$
\begin{aligned}
x_{n+1} =& x_n + \alpha(n)[h(x_n, y_n) + M_{n+1}^{(1)}], \\
y_{n+1} =& y_n + \alpha(n)[g(x_n, y_n) + M_{n+1}^{(2)}].
\end{aligned}
$$

Thus, by Theorem 4.1 in (Borkar & Meyn, 2000) or Theorem 2 in (Borkar, 2009), $\pi_n$ and $\hat{\pi}_n$ converge to equilibrium. $\qquad\square$

# B  ADDITIONAL DETAILS ON ALGORITHM

We implement our algorithm on top of MAPPO (Yu et al., 2021). MAPPO is an widely used multi-agent extension of PPO and consistently achieves strong performance on many benchmark environments. Note that our method do not include algorithm-specific structures, which means it can easily be applied to other actor-critic based algorithms, such as IPPO (de Witt et al., 2020), HATRPO (Kuba et al., 2021), MAT (Wen et al., 2022) *etc.* easily. However, while technically possible, we do not suggest a MADDPG implementation (Lowe et al., 2017) of our algorithm. This is because MADDPG provide deterministic output, but pure-strategy robust Markov perfect Perfect Bayesian equilibrium is not guaranteed to exist, as we have shown in Appedix. **??** by a counterexample.

One important thing to notice is that we empirically find using larger learning rate for adversaries during training of EIR-MAPPO do not always work well in all environments. For example, the learning dynamics of adversary in our toy environment can be unstable even with learning rate $5e - 4$, and get worse if the learning rate further increase, which is much smaller than the convergence rate suggested by previous papers (Daskalakis et al., 2020). We empirically find adversaries using slightly higher learning rates than robust agents works well, but requires extensive tuning. To tackle this problem, we maintain the central assumption of two-timescale updates (*i.e.*, updating the adversary faster and the robust agent slower), but instead update the adversary for more rollouts (denoted by `interval` in our algorithm), while update the victim for less rollouts. This do not violate our proof in Appendix A.6, since taking expectations to policy update brings the same result.

We closely follow the implementation details of MAPPO and PPO (Schulman et al., 2017), including parameter sharing, generalized advantage estimation (GAE) (Schulman et al., 2015), and other tricks in the codebase of MAPPO, available at `https://github.com/marlbenchmark/on-policy`. Note that we use fixed learning rate for both adversary and robust agents. $p_\xi(\theta|H^i)$ is a GRU (Chung et al., 2014) with input $p_\xi(b^i|o^i, h^i)$, where $h_i$ is the hidden state that summarize observations in previous timesteps and $o^i$ is observation of current timestep.

---

**Algorithm 1** *ex interim* robust c-MARL (EIR-MAPPO).

---

**Input:** Policy network of robust agents $\pi_\phi$, adversary $\hat{\pi}_{\hat{\phi}^i}$, value function $V_\psi$, belief network $p_\xi$.
**Output:** Trained policy network of robust agents $\pi_\phi$.

1: **for** k = 0, 1, 2, ... K **do**
2:    Sample $\theta \sim p(\theta)$. Initialize $\tau = []$.
3:    **for** t = 0, 1, 2, ... T **do**
4:       $\forall i \in \mathcal{N}$, perform rollout under $b_t^i = p_\xi(\theta|H_t^i)$, $\mathbf{a}_t \sim \pi_\phi(\cdot|H_t, b_t)$, $\hat{\mathbf{a}}_t \sim \hat{\pi}_{\hat{\phi}^i}(\cdot|H_t, \theta)$ and
        $s_{t+1} \sim \overline{\mathcal{P}}(s_{t+1}|s_t, \mathbf{a}_t, \theta, \hat{\pi})$, receive $H_{t+1}^i$ and $r_t$.
5:       $\tau \leftarrow \tau \cup (b_t, \mathbf{a}_t, \hat{\mathbf{a}}_t, s_{t+1}, H_{t+1}, r_t, H_t, \theta)$.
6:    **end for**
7:    **for** i = 0, 1, 2, ... $\mathcal{N}$ **do**
8:       **if** k % `interval` == 0 **then**
9:          Using $\tau$, calculate $A_\psi^i(s, \mathbf{a}, b^i)$ by GAE; calculate $b^i$ by $p_\xi$.
10:          $\phi \leftarrow \phi^i + \alpha_\phi(1 - \theta^i)\nabla \log \pi_\phi(a^i|H^i, b^i)A_\psi^i(s, \mathbf{a}, b^i)$.   `// Shared parameters`
11:          $\hat{\phi} \leftarrow \phi^i - \alpha_{\hat{\phi}}\theta^i\nabla \log \hat{\pi}_{\hat{\phi}}(a^i|H^i, \theta)A_\psi^i(s, \overline{\mathbf{a}}, b^i)$.
12:          $\psi \leftarrow \psi + \alpha_\psi\nabla_\psi(r - \gamma Q_\psi^i(s', \overline{\mathbf{a}}', b'^i) + Q_\psi^i(s, \overline{\mathbf{a}}, b^i))^2$
13:          $\xi \leftarrow \xi - \alpha_\xi\nabla_\xi(\theta \log\left(p_\xi(\theta|H^i)\right) + (1 - \theta)\log(1 - p_\xi(\theta|H^i))$
14:       **else**
15:          $\hat{\phi} \leftarrow \phi^i - \alpha_{\hat{\phi}}\theta^i\nabla \log \hat{\pi}_{\hat{\phi}}(a^i|H^i, \theta)A_\psi^i(s, \overline{\mathbf{a}}, b^i)$. `// Update adversaries for`
         `more rollouts, with others fixed. Empirically stabilize`
         `training.`
16:       **end if**
17:    **end for**
18: **end for**

---

## C   Additional Details on Experiments

### C.1   Environment Details

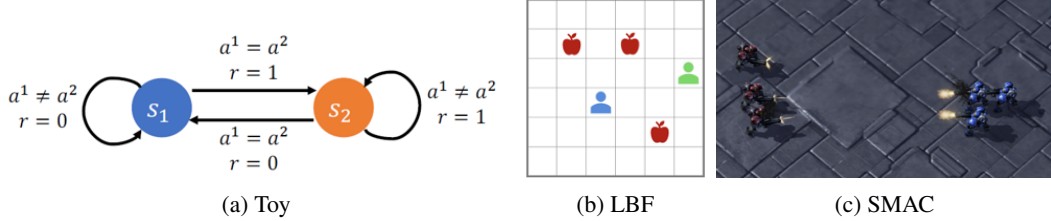

(a) Toy                     (b) LBF                (c) SMAC

Figure 7: Environments used in our experiments. The toy game is proposed by (Han et al., 2022). We use map *3m* in SMAC (Samvelyan et al., 2019) and map *12x12-4p-4f* in LBF (Papoudakis et al., 2020).

In this section, we introduce more details on environment. Again, we add the figure of environments in Fig. 7. Next, we introduce the tasks, actions and reward of each environments as follows.

**Toy environment.** The toy environment was first proposed by (Han et al., 2022) to study the effect of state-based attacks on c-MARL. In this game, two agents play simultaneously for 100 iterations to achieve maximum coordination. Specifically, in state $s_1$, two agents seeks same actions (XNOR gate), while state $s_2$ seeks two agents seeks different actions (XOR gate). During attack, the adversary can take over each agent, and perform actions to maximally attack another agent. The attack requires the robust agent to simultaneously identify other agents as adversary or allies, while taking cooperative actions if other agent is an ally, and take randomized action if other agent is an adversary.

**Level-Based Foraging environment.** Level-Based Foraging environment (Papoudakis et al., 2020) aims at a set of agents to cooperatively achieve maximum reward in food collection process. Each

agents are assigned different "levels", while the food can only be collected via agents with level higher than the food. Note that we use the *cooperative* setting in LBF, which majority agents (except the adversary) have to collaborate jointly to achieve the goal. While there do not exist a commonly used testbed in LBF, we use *12x12-4p-4f* in our experiment.

We also need to notice that, in LBF environment, an adversary is capable of physically interfere with other agents, such as blocking the way of others or intentionally colliding with other agents, resulting in a deadlock ever since. While Gleave et al. (2019) cited an important aspect of adversarial policy as "not to physically interfere with others", we turn the collision in LBF off to represent this.

**StarCraft Multi-Agent Challenge environment.** StarCraft Multi-Agent Challenge environment (Samvelyan et al., 2019) is the most commly used testbed for c-MARL, which agents control a team of red agents and seeks to win a team of blue agents with built-in AIs. We adapt the map *3m* proposed in SMAC testbed to *4m vs 3m*, with one agent as adversary. Thus, the map can still be viewed as *3m*, albeit with one adversary agents trying to fool its teammates. Note that the adversary cannot attack its allies by design of SMAC environment.

## C.2 IMPLEMENTATION DETAILS

The implementation of MAPPO, RMAAC, EAR-MAPPO and EIR-MAPPO are based on the original codebase of MAPPO (https://github.com/marlbenchmark/on-policy). The implementation of MADDPG and M3DDPG resembles the code of FACMAC (Peng et al., 2021) (https://github.com/oxwhirl/facmac) and Heterogeneous-Agent Reinforcement Learning (HARL) codebase (https://github.com/PKU-MARL/HARL). Our code are available in supplementary files, and will be open sourced after this paper is accepted.

For all environments, we set $p(\theta = \mathbf{0}_N) = 0.5$ and $p(\theta = \mathbf{1}_i) = 0.5/N$, where $\mathbf{1}_i$ denotes the one-hot vector with $\theta^i = 1$ and others 0. This probability of selecting $p(\theta)$ remains fixed throughout training process. During training, we store the model of robust agents with improved robustness without decreasing cooperation reward, evaluated on the adversary during training. While testing, we held all parameters in robust agents fixed, including policy and belief network. Then, a black-box adversary was trained following the approach of *adversarial policy*. The adversary also follows a CTDE approach, assuming assess to state, reward and local observation during training, and use local observation only in testing. For fair comparison, we attack all baselines by PPO (Schulman et al., 2017).

As for M3DDPG, note that the original version of M3DDPG (Li et al., 2019) are designed for continuous control only, where actions are continuous and can be perturbed by a small value, while in discrete control, one will have to completely change the action, or not changing the action at all. To solve that, we add the noise perturbation to the action probability of MADDPG and send it to Q function instead. We also find using large $\epsilon$ for M3DDPG will make the policy impossible to converge in fully cooperative settings: since M3DDPG add perturbations directly to each agents, resulting in an overly challenging setting. As such, we select the largest $\epsilon$ which enables maximum cooperation result in each setting.

As for RMAAC, the perturbation in training is set to $\epsilon = 0.5$ following their original paper, except for $\epsilon = 0.05$ in toy environment since otherwise the policy will not converge in normal training.

Next, we present all hyperparameters of each environment in the table below. These hyparameters follows the default in previous papers, including MAPPO (Yu et al., 2021), HARL (Zhong et al., 2023) and FACMAC (Peng et al., 2021).

Table 1: Hyperparameters for MAPPO, RMAAC, EAR-MAPPO, EIR-MAPPO in toy environment.

| Hyperparameter | Value | Hyperparameter | Value | Hyperparameter | Value |
|---|---|---|---|---|---|
| rollouts | 10 | mini-batch num | 1 | PPO epoch | 5 |
| gamma | 0.99 | max grad norm | 10 | PPO clip | 0.05 |
| gain | 0.01 | max episode len | 200 | entropy coef | 0.01 |
| actor network | MLP | actor lr | 5e-5 | eval episode | 32 |
| hidden dim | 128 | critic lr | 5e-5 | optimizer | Adam |
| belief network | GRU | adversary lr | 5e-5 | Huber loss | True |
| use PopArt | True | belief lr | 5e-5 | Huber delta | 10 |
| adversary interval | 10 | GAE lambda | 0.95 | RMAAC $\epsilon$ | 0.05 |

Table 2: Hyperparameters for MADDPG and M3DDPG in toy environment.

| Hyperparameter | Value | Hyperparameter | Value | Hyperparameter | Value |
|---|---|---|---|---|---|
| rollouts | 10 | mini-batch num | 1 | gamma | 0.99 |
| actor network | MLP | actor lr | 5e-5 | eval episode | 32 |
| hidden dim | 256 | critic lr | 5e-5 | optimizer | Adam |
| buffer size | 1000000 | batch size | 1000 | epsilon | 0.1 |

Table 3: Hyperparameters for the PPO adversary in toy environment.

| Hyperparameter | Value | Hyperparameter | Value | Hyperparameter | Value |
|---|---|---|---|---|---|
| rollouts | 10 | mini-batch num | 1 | PPO epoch | 5 |
| gamma | 0.99 | max grad norm | 10 | PPO clip | 0.05 |
| gain | 0.01 | max episode len | 200 | entropy coef | 0.01 |
| actor network | MLP | adversary lr | 5e-5 | eval episode | 32 |
| hidden dim | 128 | critic lr | 5e-5 | optimizer | Adam |
| use PopArt | True | Huber loss | True | Huber delta | 10 |
| GAE lambda | 0.95 | | | | |

Table 4: Hyperparameters for MAPPO, RMAAC, EAR-MAPPO, EIR-MAPPO in SMAC environment.

| Hyperparameter | Value | Hyperparameter | Value | Hyperparameter | Value |
|---|---|---|---|---|---|
| rollouts | 20 | mini-batch num | 1 | PPO epoch | 5 |
| gamma | 0.95 | max grad norm | 10 | PPO clip | 0.05 |
| gain | 0.01 | max episode len | 200 | entropy coef | 0.01 |
| actor network | MLP | actor lr | 5e-4 | eval episode | 32 |
| hidden dim | 128 | critic lr | 5e-4 | optimizer | Adam |
| belief network | GRU | adversary lr | 5e-4 | Huber loss | True |
| use PopArt | True | belief lr | 5e-4 | Huber delta | 10 |
| adversary interval | 5 | GAE lambda | 0.95 | RMAAC $\epsilon$ | 0.05 |

Table 5: Hyperparameters for MADDPG and M3DDPG in SMAC environment.

| Hyperparameter | Value | Hyperparameter | Value | Hyperparameter | Value |
|---|---|---|---|---|---|
| rollouts | 20 | mini-batch num | 1 | gamma | 0.99 |
| actor network | MLP | actor lr | 5e-4 | eval episode | 32 |
| hidden dim | 256 | critic lr | 5e-4 | optimizer | Adam |
| buffer size | 1000000 | batch size | 1000 | epsilon | 0.01 |

Table 6: Hyperparameters for the PPO adversary in SMAC environment.

| Hyperparameter | Value | Hyperparameter | Value | Hyperparameter | Value |
|---|---|---|---|---|---|
| rollouts | 20 | mini-batch num | 1 | PPO epoch | 5 |
| gamma | 0.99 | max grad norm | 10 | PPO clip | 0.05 |
| gain | 0.01 | max episode len | 200 | entropy coef | 0.01 |
| actor network | MLP | adversary lr | 5e-4 | eval episode | 32 |
| hidden dim | 128 | critic lr | 5e-4 | optimizer | Adam |
| use PopArt | True | Huber loss | True | Huber delta | 10 |
| GAE lambda | 0.95 | | | | |

Table 7: Hyperparameters for MAPPO, RMAAC, EAR-MAPPO, EIR-MAPPO in LBF environment.

| Hyperparameter | Value | Hyperparameter | Value | Hyperparameter | Value |
|---|---|---|---|---|---|
| rollouts | 20 | mini-batch num | 1 | PPO epoch | 5 |
| gamma | 0.99 | max grad norm | 10 | PPO clip | 0.05 |
| gain | 0.01 | max episode len | 200 | entropy coef | 0.01 |
| actor network | MLP | actor lr | 5e-4 | eval episode | 32 |
| hidden dim | 128 | critic lr | 5e-4 | optimizer | Adam |
| belief network | GRU | adversary lr | 5e-4 | Huber loss | True |
| use PopArt | True | belief lr | 5e-4 | Huber delta | 10 |
| adversary interval | 5 | GAE lambda | 0.95 | RMAAC $\epsilon$ | 0.05 |

Table 8: Hyperparameters for MADDPG and M3DDPG in LBF environment.

| Hyperparameter | Value | Hyperparameter | Value | Hyperparameter | Value |
|---|---|---|---|---|---|
| rollouts | 20 | mini-batch num | 1 | gamma | 0.99 |
| actor network | MLP | actor lr | 5e-5 | eval episode | 32 |
| hidden dim | 256 | critic lr | 5e-5 | optimizer | Adam |
| buffer size | 1000000 | batch size | 1000 | epsilon | 0.1 |

Table 9: Hyperparameters for the PPO adversary in LBF environment.

| Hyperparameter | Value | Hyperparameter | Value | Hyperparameter | Value |
|---|---|---|---|---|---|
| rollouts | 20 | mini-batch num | 1 | PPO epoch | 5 |
| gamma | 0.99 | max grad norm | 10 | PPO clip | 0.05 |
| gain | 0.01 | max episode len | 200 | entropy coef | 0.01 |
| actor network | MLP | adversary lr | 5e-4 | eval episode | 32 |
| hidden dim | 128 | critic lr | 5e-4 | optimizer | Adam |
| use PopArt | True | Huber loss | True | Huber delta | 10 |
| GAE lambda | 0.95 | | | | |

