# OpenReview forum: "Byzantine Robust Cooperative Multi-Agent Reinforcement Learning as a Bayesian Game"
_ICLR.cc/2024/Conference — ICLR 2024 poster_

### Official Review · Reviewer_UzHb · 2023-10-23

**Soundness:** 3 good
**Presentation:** 2 fair
**Contribution:** 3 good
**Rating:** 8
**Confidence:** 4

**Summary:**

The paper addresses Byzantine failures in multi-agent reinforcement learning (MARL), where agents can adopt malicious behavior due to malfunction caused by hardware/software faults. First, a new problem setting, called Bayesian Adversarial Robust Dec-POMDP (BARDec-POMDP), is proposed to model adversarial intervention as a transition, where originally cooperative joint actions are modified before being executed in the actual environment. The setting is inspired by Bayesian games, where agents have types that need to be inferred by other agents through their beliefs. The goal is to learn a more fine-grained solution than prior work, where policies are able to collaborate with functional agents, while being robust against adversaries therefore finding a better cooperation-robustness trade-off. Ex post mixed-strategy robust Bayesian Markov perfect equilibrium is proposed as a solution concept for BARDec-POMDPs, which is claimed to be weakly dominant over equilibria that were pursued in previous works.

Based on this setting, a two-timescale actor-critic is proposed based on the robust Harsanyi-Bellman equation, defining the maximin value according to the ideal case (before any adversarial modification) and the adversarial case. The policy gradient uses the value of the adversarial case as critic. The approach is evaluated in a variety of games, including a matrix game, a gridworld game, and a map of SMAC. It is also evaluated against different attacker types.

**Strengths:**

The paper addresses a very important problem, which is often neglected in the MARL community.

The proposed setting (BARDec-POMDP) can model realistic scenarios, where agents can behave adversarially due to malfunctioning.

While most works on robust MARL focus on defenses against any kind of adversarial attack, which leads to conservative policies, the paper aims to find a trade-off between collaborating with actual functional agents and robustness against actual adversaries, which is a more desirable goal for real-world applications. Bayesian games provide a neat framework to discern between these types of agents, which is a convincing proposal.

The solution concept seems valid.

I like the evaluation, which first starts with a small toy problem before scaling up to larger benchmarks like SMAC. I also appreciate the evaluation with different adversarial strategies.

**Weaknesses:**

My main concern with the paper is the lack of self-containment: The paper excessively references the appendix, which hurts readability (it is not convenient to peek into the appendix for almost every paragraph of the main paper).

The paper needs a clear prioritization of what is really important and what can be safely left in the appendix. For example, important proofs can be sketched informally with little space, but delegating them to the appendix completely is not acceptable.

Before checking, if I can raise my score, I need a revised version of the paper that is more self-contained, e.g., where the number of appendix references is significantly reduced and proofs are at least sketched such that the reader is not forced to switch documents all the time.

**Minor Comments**
- In Dec-POMDPs (Section 2.1), policies select their actions by the history of past observations and actions. Conditioning only on observations is insufficient since they are not Markovian. The definition in Section 2.2 is correct, though.
- Line 119: “was replaced” - “**is** replaced”
- Line 277: “four type of threats” - “four type**s** of threats”
- The text size in Figure 2 is too small for printed versions.

**Questions:**

1. Why do adversaries condition on observations despite them not being Markovian (in contrast to the non-adversarial policies)? Is this intentional or a mistake?
2. Do you have any intuition on why M3DDPG performs so poorly in the presence of an adversary compared to the standard MAPPO without any robustness mechanism in Figure 4?

---

> ### Author Response · Authors · 2023-11-17
> **Response of Submission 872 to Reviewer UzHb**
>
> We thank Reviewer UzHb for recognizing the significance, problem setting, theoretical concept, solution concept and evaluation of our paper. A detailed response is provided below.
>
> > Q1: Self-containment and conciseness.
>
> We appreciate this valuable feedback. To address this, we have:
>
> * Removed unnecessary discussions, reducing references to the appendix from 17 times to 8 times and decreasing the appendix length from 21 pages to 12 pages. We have meticulously revised the manuscript to ensure it remains focused on the core message. For completeness, we will host a GitHub page for additional experiment results upon acceptance.
>
> * Included a concise proof sketch for each theorem to enhance self-containment.
>
> * Reduced the use of mathematical notations and added more textual explanations.
>
> Please refer to our global response and the revised manuscript for more details.
>
> > Q2: Why do adversaries condition on observations despite them not being Markovian (in contrast to the non-adversarial policies)? Is this intentional or a mistake?
>
> Thank you for pointing out the concern. Please allow us to clarify that in our original draft, for both defender and adversary, their policies are conditioned on histories $H$, not observations $o$. For example:
>
> * Section. 2.1, paragraph 2, "...each agent $i$ selects its action $a_t^i \in \mathcal A^i$ using its policy $\pi^i(\cdot|H_i^t)$"; "the goal of all agents is to learn a joint policy ... that maximize the long-term return "$J(\mathbf{\pi}) = \mathbb{E}\left[\sum_{t=0}^{\infty} \gamma^t r_t|s_0, \mathbf{a}_t \sim \pi(\cdot|H_t)\right]$".
> * Section. 2.2, paragraph 2, "replaced by action $\hat{a}^i_t$ sampled from an adversary with policy $\hat{\pi}^i(\cdot|H_t^i, \theta)$"; "the value function can be defined as $V_\theta(s) = \mathbb{E}\left[\sum_{t=0}^{\infty} \gamma^t r_t|s_0=s, \mathbf{a}_t \sim \pi(\cdot|H_t), \hat{\mathbf{a}}_t \sim \hat{\pi}(\cdot|H_t, \theta)\right]$".
>
> A thorough review of our paper confirms that the policy of both defenders and adversaries rely on histories, not observations.
>
> > Q3: Do you have any intuition on why M3DDPG performs so poorly in the presence of an adversary compared to the standard MAPPO without any robustness mechanism in Figure 4?
>
> There are two key reasons for M3DDPG’s underperformance in adversarial contexts, as compared to standard MAPPO without robustness mechanisms.
>
> First, as discussed in Proposition 2.2, the existence of a pure-strategy equilibrium (RMPBE) isn't guaranteed. M3DDPG, which derives from MADDPG, inherently generates deterministic policies. Such policies may not be optimal in robust cooperative MARL contexts where a pure-strategy equilibrium is not assured. Despite this, M3DDPG was included as a baseline due to its recognition in the field.
>
> Second, as a part of the max entropy RL family, MAPPO employs stochastic policies that encourage diverse solution strategies, thereby enhancing resilience to perturbations. This advantage of max entropy policies is well-documented in several studies [1, 2, 3, 4].
>
> [1] Ziebart, Brian D. Modeling purposeful adaptive behavior with the principle of maximum causal entropy. Ph.D. Thesis, Carnegie Mellon University, 2010.
>
> [2] Haarnoja, Tuomas, et al. "Reinforcement learning with deep energy-based policies." ICML 2017.
>
> [3] Haarnoja, Tuomas, et al. "Soft actor-critic: Off-policy maximum entropy deep reinforcement learning with a stochastic actor." ICML 2018.
>
> [4] Eysenbach, Benjamin, and Sergey Levine. "Maximum entropy rl (provably) solves some robust rl problems." ICLR 2022.
>
> > Minors:
>
> Thank you for your suggestions. We have meticulously revised the manuscript to correct grammatical errors and enhanced the clarity of Figure 2 in our revised submission.

---

> > ### Author Response · Authors · 2023-11-20
> > **Invitation to Participate in Reviewer-Author Discussion**
> >
> > Dear Reviewer UzHb,
> >
> > Thanks for the insightful comments and valuable advice on our paper. Your constructive feedback are vital to the enhancement of our work.
> >
> > As the reviewer-author discussion phase is going to end, we would like to invite you for further discussion of our paper and are happy to response to your further questions. Your continued input would be of immense significance to us, and we deeply appreciate your consideration in dedicating further time to our paper.

---

> ### Comment · Reviewer_UzHb · 2023-11-21
> **Follow-Up**
>
> Thank you for the rebuttal and the revised version. Since I asked for a revision that affected the whole paper, I needed to completely re-read the new version to check.
>
> The revision reads well to me now. Therefore, I am happy to increase my score since all my concerns have been sufficiently addressed.

---

> > ### Author Response · Authors · 2023-11-21
> > **Official Comments by Authors**
> >
> > Dear Reviewer UzHb,
> >
> > We really appreciate your prompt feedback and the amount of time you have spent reviewing our paper! We sincerely thank you for your valuable comments and suggestions. The paper's quality has been greatly enhanced by your tremendous efforts.
> >
> > Appreciated!
> >
> > Best regards,
> >
> > Authors of submission 872

---

### Official Review · Reviewer_EssV · 2023-11-07

**Soundness:** 3 good
**Presentation:** 2 fair
**Contribution:** 3 good
**Rating:** 5
**Confidence:** 3

**Summary:**

The paper addresses the robustness of multi-agent reinforcement learning against Byzantine failures. A Bayesian Adversarial Robust
Dec-POMDP (BARDec-POMDP) framework is proposed.  The theoretical formulation of the problem is one of the contributions of the
paper and an actor-critic algorithm that produces convergence under some conditions.  Convergence is not ensured but experimentally
the approach shows great resilience against a spectrum of adversaries.

The paper makes some strong assumptions, like the fact that in each episode there is only one attacker. Other assumptions are consistency and sequential rationality, which are used to form an ex interim robust Markov perfect Bayesian equilibrium (RMPBE). With some additional assumptions, the existence of ex ante and ex interim RMPBE are guaranteed. For the ex interim equilibrium, the Q function can be written using the Bellman-type equation for the two Q functions, which the paper calls "the robust Harsanyi-Bellman equation." With a few more assumptions, updating the value function via the Bayes rule guarantees convergence to the optimal value of Q.

The algorithm proposed is applied to different problems, a toy metrics game and some map problems. For the attacks, four types of threats are considered: non-oblivious adversaries, random agents, noisy observations, and transferred adversaries. The experimental results cover robustness over non-oblivious attacks and various types of attacks.

**Strengths:**

The novelty of the paper is in the theoretical formulation of the problem and in the actor-critic algorithm that produces convergence
under some conditions.  Convergence is not ensured but experimentally the approach shows great resilience against a spectrum of adversaries.

**Weaknesses:**

- The fact that convergence cannot be guaranteed despite the assumptions made is a weakness of the paper. The method proposed is complex, and the assumptions are strong, yet there is no guarantee of convergence
- The proofs are all in the Appendix, which makes the paper quite long, too long for a conference paper, and that makes it appear incomplete.  Even the analysis of two of the environments used for the experiments is in the Appendix. Expecting the reviewers to read a paper of 36 pages is too much for a conference.  Without the Appendices the paper is incomplete.
- The paper is hard to read, it assumes significant knowledge of the field and is full of acronyms and citations that break the flow of the text.

After the discussion and the changes made to the paper, I reduced my criticism about convergence and increased the rating I gave for contribution. However, the presentation needs more work.  Many of the corrections made to the paper have grammar errors, like mismatches of singular/plural. I understand the rush of making the changes, but the writing is not where it should be.  I also still object to the length of the paper.

**Questions:**

If you were to write a paper that fits in the page limits, would you include some of the material from the Appendices and drop some of the material currently in the paper or leave the paper unchanged?  I am wondering what you think are the most important parts needed to present the work without exceeding the page limits.

---

> ### Author Response · Authors · 2023-11-17
> **Response of Submission 872 to Reviewer EssV**
>
> We thank Reviewer EssV for appreciating the theoretical formulation and empirical evaluation for our paper. A detailed response is provided below.
>
> > Q1: Convergence cannot be guaranteed.
>
> Thank you for pointing out the concern. Firstly, please allow us to clarify that we have established convergence in our original draft. As stated on page 6, Section 3.2, "Using common assumptions in stochastic approximation proposed by Borkar (Borkar, 1997; Borkar & Meyn, 2000; Borkar, 2009), we show that Theorem 3.1 converges almost surely to ex interim robust Markov perfect Bayesian equilibrium, with detailed assumptions and proof deferred to Appendix. A.8". This proof leverages stochastic approximation theory and is underscored in both our abstract and introduction.
>
> Secondly, the term "almost sure" convergence implies that the sequence of defender and attacker policies converge with probability 1 as the number of iterations goes to infinity.
>
> Thirdly, for greater clarity, we have added a bolded subtitle "Convergence" in Section 3.2 and restructured the relevant paragraph.
>
> Fourthly, while our approach and methodology indeed rely on specific assumptions and mathematical techniques, the implementation of our algorithm is practical and straightforward. It involves conditioning the policy and value function on the belief of the current type, combined with a two-timescale update strategy (i.e., faster policy updates for the adversary and slower for the defender).
>
> > Q2: The paper is too long, and reference appendix for too many times. Concentrate on the most important parts.
>
> We appreciate this valuable feedback. To address this, we have:
>
> * Removed unnecessary discussions, reducing references to the appendix from 17 times to 8 times and decreasing the appendix length from 21 pages to 12 pages. We have meticulously revised the manuscript to ensure it remains focused on the core message. For completeness, we will host a GitHub page for additional experiment results upon acceptance.
>
> * Included a concise proof sketch for each theorem to enhance self-containment.
>
> * Reduced the use of mathematical notations and added more textual explanations.
>
> Please refer to our global response and the revised manuscript for more details.
>
> > Q3: The paper is hard to read, it assumes significant knowledge of the field and is full of acronyms and citations that break the flow of the text.
>
> Thank you for this suggestion. In our revised version, we have carefully go through the whole paper and reduced the use of mathematical notations and added more textual explanations. However, it's important to note that certain expressions equations cannot be overly simplified for correctness.

---

> > ### Author Response · Authors · 2023-11-20
> > **Invitation to Participate in Reviewer-Author Discussion**
> >
> > Dear Reviewer EssV,
> >
> > Thanks for the insightful comments and valuable advice on our paper. Your constructive feedback are vital to the enhancement of our work.
> >
> > As the reviewer-author discussion phase is going to end, we would like to invite you for further discussion of our paper and are happy to response to your further questions. Your continued input would be of immense significance to us, and we deeply appreciate your consideration in dedicating further time to our paper.

---

> ### Author Response · Authors · 2023-11-21
> **Follow-Up: Reviewer-Author Discussion Period Ends Tomorrow**
>
> Dear Reviewer EssV,
>
> We sincerely thank you for raising the score of our paper, the paper's quality has been greatly enhanced by your tremendous efforts. As the reviewer-author discussion period ends tomorrow, we hope that our recent updates and responses have adequately addressed your queries and concerns. Should you have any further questions, we are committed to promptly addressing them within the remaining timeframe.
>
> Best regards,
> Authors of Submission 872

---

### Official Review · Reviewer_FinH · 2023-11-07

**Soundness:** 3 good
**Presentation:** 3 good
**Contribution:** 3 good
**Rating:** 6
**Confidence:** 4

**Summary:**

The research explores the stability of cooperative multi-agent reinforcement learning when faced with adversarial strategies and disruptions. It introduces the Bayesian Adversarial Robust Decentralized Partially Observable Markov Decision Process (BARDec-POMDP) as a framework for addressing these challenges. The study demonstrates that conventional robust cooperative MARL approaches only achieve an ex interim equilibrium and advances the concept of an ex post equilibrium for BARDec-POMDP, which it argues is superior under worst-case scenarios. To attain this ex post equilibrium, the study presents a novel actor-critic method named EPR-MAPPO. Testing on simple games, the LBF environment, and the StarCraft Multi-Agent Challenge (SMAC) illustrates that EPR-MAPPO outperforms standard models, showcasing resilience against four distinct threat categories.

**Strengths:**

The manuscript is easy to follow. Addressing the robustness of Multi-Agent Reinforcement Learning (MARL) is crucial, particularly for its practical deployment in real-life situations. The introduction of the ex post equilibrium within the BARDec-POMDP framework by the authors offers a thoughtful contribution to the field. The paper includes an in-depth examination of the algorithms it introduces. Furthermore, the empirical findings presented are coherent and compelling.

**Weaknesses:**

The document frequently refers to its appendix, suggesting that it does not stand alone as effectively as it could. To enhance the feeling of a complete and self-contained work, the authors should further take simplification, such as using simpler terms within the equations for greater clarity.

**Questions:**

How can the proposed method benefit real-world applications? Any concrete examples?

**Details Of Ethics Concerns:**

Non

---

> ### Author Response · Authors · 2023-11-17
> **Response of Submission 872 to Reviewer FinH**
>
> We thank Reviewer FinH for recognizing the significance, clarity, theoretical contribution and empirical findings of our paper. A detailed response is provided below.
>
> > Q1: The document frequently refers to its appendix, suggesting that it does not stand alone as effectively as it could. The authors should further take simplification, such as using simpler terms within the equations for greater clarity.
>
> We appreciate this valuable feedback. To address this, we have:
>
> * Removed unnecessary discussions, reducing references to the appendix from 17 times to 8 times and decreasing the appendix length from 21 pages to 12 pages. We have meticulously revised the manuscript to ensure it remains focused on the core message. For completeness, we will host a GitHub page for additional experiment results upon acceptance.
>
> * Included a concise proof sketch for each theorem to enhance self-containment.
>
> * Reduced the use of mathematical notations and added more textual explanations.
>
> Please refer to our global response and the revised manuscript for more details.
>
> > Q2: How can the proposed method benefit real-world applications? Any concrete examples?
>
> Let's illustrate this by examples. In robot swarm control [1, 2], individual robots may behave unpredictably due to malfunctions or external interference [3, 4]. Our approach studies the lower bound for these scenarios where a swarm faces a worst-case, non-oblivious adversary, as discussed in our introduction.
>
> Apart from robot swarms, in traffic light control [5], individual lights might malfunction or be compromised, leading to a non-cooperative system. Similarly, in power grid control [6], nodes might fail or transmit erroneous signals. Our cooperative MARL approach is designed to adaptively respond to these challenges, ensuring efficient operation of these systems during deployment. We plan to apply our algorithm in these domains as future work, as discussed in our conclusion section.
>
> [1] Hüttenrauch, Maximilian, Sosic Adrian, and Gerhard Neumann. "Deep reinforcement learning for swarm systems." JMLR 2019.
>
> [2] Batra, Sumeet, et al. "Decentralized control of quadrotor swarms with end-to-end deep reinforcement learning." CoRL 2022.
>
> [3] Giray, Sait Murat. "Anatomy of unmanned aerial vehicle hijacking with signal spoofing." RAST 2013.
>
> [4] Ly, Bora, and Romny Ly. "Cybersecurity in unmanned aerial vehicles (UAVs)." Journal of Cyber Security Technology 2021.
>
> [5] Chu, Tianshu, et al. "Multi-agent deep reinforcement learning for large-scale traffic signal control." IEEE TITS 2019.
>
> [6] Xi, Lei, et al. "Smart generation control based on multi-agent reinforcement learning with the idea of the time tunnel." Energy 2018.

---

> > ### Author Response · Authors · 2023-11-20
> > **Invitation to Participate in Reviewer-Author Discussion**
> >
> > Dear Reviewer FinH,
> >
> > Thanks for the insightful comments and valuable advice on our paper. Your constructive feedback are vital to the enhancement of our work.
> >
> > As the reviewer-author discussion phase is going to end, we would like to invite you for further discussion of our paper and are happy to response to your further questions. Your continued input would be of immense significance to us, and we deeply appreciate your consideration in dedicating further time to our paper.

---

> > ### Author Response · Authors · 2023-11-21
> > **Follow-Up: Reviewer-Author Discussion Period Ends Tomorrow**
> >
> > Dear Reviewer FinH,
> >
> > We are grateful for your insightful feedback on our paper. As the reviewer-author discussion period ends tomorrow, we hope that our recent updates and responses have adequately addressed your concerns. Should you have any further questions, we are committed to promptly addressing them within the remaining timeframe.
> >
> > Warm regards,
> > Authors of Submission 872

---

> > > ### Comment · Reviewer_FinH · 2023-11-22
> > > **Thanks for the response**
> > >
> > > Thanks for the efforts to revise the paper and answering my questions. It looks better now. However, I agree with Reviewer Essa that the paper still needs more works on its presentation. Therefore, I would like to maintain the current rating.

---

> > > > ### Author Response · Authors · 2023-11-23
> > > > **Thanks for the feedback**
> > > >
> > > > Dear Reviewer FinH,
> > > >
> > > > We really appreciate your prompt feedback and the amount of time you have spent reviewing our paper! We sincerely thank you for your valuable comments and suggestions. The paper's quality has been greatly enhanced by your tremendous efforts. Inspired by your support, we are committed to further enhancing our paper's presentation upon acceptance.
> > > >
> > > > Warm regards,
> > > >
> > > > Authors of Submission 872

---

### Author Response · Authors · 2023-11-17
**Paper Rebuttal Summary**

We express our gratitude to the reviewers for acknowledging the significance (FinH, UzHb), clarity (FinH), theoretical formulation (FinH, EssV, UzHb), and empirical evaluation (FinH, EssV, UzHb) of our paper. We also appreciate their insightful questions and suggestions. The revisions are highlighted in blue.

Our research systematically explores action uncertainties in cooperative Multi-Agent Reinforcement Learning (MARL), distinguishing itself by considering uncertainties and incomplete information associated with adversaries. Our results yield a marked improvement in robustness across various action uncertainties, suggesting a promising approach to mitigate potential failures in cooperative MARL and potentially hastening real-world applications.

The major concern raised by reviewers was the frequent references to the appendix, impacting the paper's self-containment. To address this without altering our main contributions, we have made the following revisions:

* [Adding proof sketch]: We now include a proof sketch for each proposition in the main text, enabling readers to grasp our proof techniques without needing to consult the appendix.

* [Reducing the amount of math]: We have replaced some mathematical notations with text descriptions for enhanced readability.

* [Reorganize the appendix]: Based on reviewers' feedback, we have eliminated derivations, discussions, and experiments unrelated to our core findings. This cuts down references to the appendix from 17 times to 8 times and reduces appendix length from 21 pages to 12 pages. The revised manuscript maintains its core messages. For completeness, we will host a GitHub page for additional experiment results upon acceptance.

Specific deletions include:

1. Removing the non-essential counterexample for the non-existence of pure-strategy RMPBE (Appendix A.3), as it's common knowledge that zero-sum games are not guaranteed to have a pure-strategy equilibrium.

2. Omitting the robust Harsanyi-Bellman equation for V function (Appendix A.5), as it's not used in our algorithms.

3. Reformulating discussions on the convergence of our actor-critic algorithm and adding "convergence" as a subtitle for clarity.

4. Deleting discussions on policy gradient considering opponent awareness, adversary considerations (Appendix A.9) intended for future research.

5. Consolidating experiment details into a single appendix reference.

6. Removing additional experimental results and analyses that are redundant to the main text (Appendix C.3-C.6). For completeness, we will host a GitHub page for these results upon acceptance.

7. Summarizing concluding remarks (Appendix D) in conclusions and delete it in appendix.

These revisions ensure our paper remains comprehensive yet self-contained, aligning with the reviewers' guidance. Responses to other specific questions raised by the reviewers have been addressed individually.

---

### Meta-Review · Area_Chair_Y37b · 2023-12-06

**Metareview:**

This paper has studied the problem of multi-agent reinforcement learning in the face of adversarial strategies and disruptions. To this end, it introduces a Bayesian framework, shows the drawback of conventional robust cooperative MARL approaches, and develops a novel actor-critic method to overcome the drawback. The paper is well-written, and the motivation is clear. The theoretical parts are solid (though some reviewers have the problem of deferring the proof to the appendix). I would recommend the authors to incorporate all the feedback from the reviewers in the final version.

**Justification For Why Not Higher Score:**

It is a paper that the reviewers have some concerns regarding the presentation, and the assumptions used for the theoretical results. The organization and self-containedness of the paper definitely need a bit more work.

**Justification For Why Not Lower Score:**

It is a good paper that studies an important problem in multi-agent RL, and the results are mostly novel.

---

### Decision · Program_Chairs · 2024-01-16

Accept (poster)